# Macrophage invasion into the *Drosophila* brain requires JAK/STAT-dependent MMP activation in the blood–brain barrier

**Bente Winkler** ⓘ**, Dominik Funke, Christian Klämbt** ⓘ*

Institut für Neuro- und Verhaltensbiologie, Universität Münster, Münster, Germany

* klaembt@uni-muenster.de

## Abstract

The central nervous system is well-separated from external influences by the blood–brain barrier. Upon surveillance, infection or neuroinflammation, however, peripheral immune cells can enter the brain where they often cause detrimental effects. To invade the brain, immune cells not only have to breach cellular barriers, but they also need to traverse associated extracellular matrix barriers. Neither in vertebrates nor in invertebrates is it fully understood how these processes are molecularly controlled. We recently established *Drosophila melanogaster* as a model to elucidate peripheral immune cell invasion into the brain. Here, we show that neuroinflammation leads to the expression of Unpaired cytokines that activate the JAK/STAT signaling pathway in glial cells of the blood–brain barrier. This in turn triggers the expression of matrix metalloproteinases enabling remodeling of the extracellular matrix enclosing the fly brain and a subsequent invasion of immune cells into the brain. Our study demonstrates conserved mechanisms underlying immune cell invasion of the nervous system in invertebrates and vertebrates and could, thus, further contribute to understanding of JAK/STAT signaling during neuroinflammation.

## Introduction

The central nervous system (CNS) is a unique organ computing sensory input to orchestrate appropriate responses of the animal. The CNS is formed by glial cells and neurons that require a constant environment in order to reliably calculate information. Therefore, the CNS is protected from circulating factors by the blood–brain barrier (BBB), which not only ensures proper ion and solute homeostasis, but also restricts pathogen and immune cell entry into the CNS.

In vertebrates, the BBB is established by endothelial cells which form complex tight junctions to prevent paracellular diffusion [1]. Moreover, endothelial as well as pericyte cells produce an endothelial basement membrane (BM) composed of α4 and α5 laminins, type IV collagen, nidogen 1 and the heparan sulfate proteoglycan, perlecan [2–4]. The blood vessels of the CNS are, in addition, separated from the CNS parenchyma by a second BM that is formed by astrocytic endfeet that face the vessels. At the level of capillaries and postcapillaries venules, this second BM and the astrocyte endfeet layer contribute to barrier function of the CNS blood vessels [5]. This parenchymal BM is molecularly distinct from the endothelial BM by

**Data availability statement:** All relevant data are within the paper and its Supporting information files.

**Funding:** This work was supported by a grant of the German research foundation (DFG, SFB 1009 A4; https://gepris-extern.dfg.de/gepris/projekt/194468054) to C.K. The funders had no role in study design, data collection and analysis, decision to publish, or preparation of the manuscript.

**Competing interests:** The authors have declared that no competing interests exist.

**Abbreviations :** AMPs, antimicrobial peptides; APF, after puparium formation; BBB, blood–brain barrier; BM, basement membrane; CNS, central nervous system; CTCF, corrected total cell fluorescence; ECM, extracellular matrix; Imd, Immune deficiency; JAK/STAT, Janus kinase/signal transducer and activator of transcription; JNK, c-Jun N-terminal kinase; MAPK, mitogen-activated protein kinase; MMPs, matrix metalloproteinases; NF-κB, nuclear factor-κB; PDGF/VEGF, platelet-derived growth factor/vascular endothelial growth factor; PGs, perineurial glial cells; PGRPs, peptidoglycan recognition proteins; Puc, puckered; qRT-PCR, quantitative real-time PCR; SPGs, subperineurial glial cells; TIMP, tissue inhibitor of metalloproteases; Upd, unpaired; VCAM-1, vascular cell adhesion protein 1.

the expression of α1 and α2 laminin [6]. The parenchymal BM and astrocytic endfeet form the glia limitans located basal to the endothelial BM [7–9].

Within the vertebrate CNS, immune surveillance is mainly performed by microglial cells which derive from the yolk sack and migrate into the CNS during embryonic development [10]. They can affect neuronal survival by phagocytosis of cellular debris and release of soluble factors including reactive oxygen species and nerve growth factors [11]. In addition to local microglial cells, resident perivascular macrophages located between the parenchymal and endothelial/smooth muscle BMs as well as peripheral immune cells can enter the CNS over the BBB during immune surveillance and inflammation [12]. However, invading immune cells can also cause the development of neuroinflammation. In a model for multiple sclerosis, experimental autoimmune encephalomyelitis, disease symptoms only appear when peripheral immune cells overcome the parenchymal BM of the glia limitans [9]. This process is mainly mediated by an upregulation of the *matrix metalloproteinases* (*MMPs*) MMP-2 and MMP-9 in astrocytes. They regulate the activity of different cytokines and cleave the parenchymal BM in order to allow the immune cell invasion into the parenchyma [5,9]. Among the pathways that can up-regulate MMP expression are Janus kinase/signal transducer and activator of transcription (JAK/STAT), c-Jun N-terminal kinase (JNK) and the nuclear factor-κB (NF-κB) signaling pathways [13–15]. However, the molecular mechanisms leading to MMP-2 and MMP-9 expression by astrocytes at the parenchymal border are not completely understood [5].

In invertebrates like *Drosophila melanogaster* innate immunity is evolutionary conserved, and thus flies may provide a good model to study immune responses in the CNS. The *Drosophila* CNS is protected by a specific BBB which, as in primitive vertebrates, is solely established by glial cells and an additional extracellular matrix (ECM) layer, the neural lamella [16–19]. The BBB forming glial cells are the subperineurial glial cells (SPGs) and perineurial glial cells (PGs). The SPGs form occluding septate junctions that efficiently block paracellular diffusion, and the infiltration of pathogens and peripheral immune cells [20]. The PGs participate in the secretion of the neural lamella and are involved in the metabolic supply of the CNS [21]. No microglia-like cells are found in invertebrates; however, local glial cells like ensheathing glia are able to respond to injuries by phagocytosis of cellular debris [22,23].

The cellular immune response in *Drosophila* is mediated by hemocytes, primary systemic immune cells most of which are migratory phagocytes or macrophages [24]. By expression of different receptors like peptidoglycan recognition proteins (PGRPs) they are able react to bacterial infection within the organism [25–27]. Especially during pupal stages macrophages are highly migratory, can relocate by chemotaxis and show a high diversity [28].

The main pathways underlying the humoral immune response are based on conserved NF-κB signaling pathways. They are subdivided into the Toll and the Immune deficiency (Imd) pathway which are activated by gram-positive bacteria or fungi, or by gram-negative bacteria, respectively [29]. Activation of either of these pathways results in the translocation of a NF-κB-type transcription factor into the nucleus promoting the expression of antimicrobial peptides (AMPs) which in turn directly target the respective pathogens. Specifically, the Imd pathway can be activated by binding of cell-wall components of pathogenic bacteria to the PGRPs, PGRP-LC, PGRP-SD or PGRP-LE [25,26,30,31]. PGRP activation leads to the cleavage of the inhibitory domain of the NF-κB factor Relish by a Death-related ced-3/Nedd2-like caspase (Dredd). This then allows its translocation into the nucleus and the subsequent activation of transcription of AMPs [29].

Moreover, the Imd pathway can also bifurcate into the JNK pathway and thereby activate it via the TGF-β activated kinase 1 (Tak1) [32]. The JNK pathway is part of the mitogen-activated protein kinase (MAPK) pathway and its stimulation results in activation of the

transcription factor AP-1 [33]. Activation of the JNK pathway can be detected using a *puckered* (*puc*) reporter and results in the expression of MMPs and Unpaired (Upd) family cytokines [14,34–37].

Three different Upd cytokines, Upd1, Upd2 and Upd3 activate the conserved JAK/STAT pathway by binding to the Domeless (Dome) receptor, the only known *Drosophila* Upd receptor [38]. Dome activation triggers recruitment of the Janus kinase Hopscotch (Hop) which in turn phosphorylates the transcription factor STAT92E inducing its dimerization, and translocation into the nucleus [39,40]. Among others, the JAK/STAT pathway is implicated in the systemic immune response to epidermal wounds, tumors, mechanical stress and parasitoid wasp infection [41].

Recently, we developed a neuroinflammation model in *Drosophila* to study how immune responses trigger macrophage invasion into the CNS [20]. The activation of the Imd pathway by panglial expression of *PGRP-LE* results in neuroinflammation and triggers the invasion of peripheral macrophages into the CNS during pupal stages. The invasion of macrophages depends on upregulation of the *platelet-derived growth factor/vascular endothelial growth factor (PDGF/VEGF)-related factor Pvf2*. Here, we demonstrate that panglial immunity induction also triggers a remodeling of the neural lamella during pupal stages, which favors immune cell invasion. Concomitantly, it induces an upregulation of Upd ligands. BBB glial cells respond by activation of JAK/STAT signaling and activate expression of MMPs, of which only two are encoded by the *Drosophila* genome [42]. Ectopic expression of *Mmp1* or *Mmp2* within the BBB triggers macrophage invasion into the CNS. Our study demonstrates upregulation of MMP expression at the BBB and subsequent remodeling of neural lamella enabling macrophage invasion into the *Drosophila* CNS upon neuroinflammation and may be relevant to neuroinflammation in vertebrates.

## Results

### Panglial immunity induction triggers remodeling of the neural lamella

The *Drosophila* nervous system is covered by a dense ECM layer called neural lamella (S1 Fig). During development, glial cells of the BBB as well as macrophages deposit different ECM components which are then assembled into a dense neural lamella. Disturbance of a proper neural lamella formation disrupts ventral nerve cord condensation in embryonic stages [43, 44]. During larval stages, the ECM is thought to provide a stiff counter bearing required during remodeling of the CNS [45,46]. Therefore, the neural lamella needs to be constantly remodeled.

Upon a panglial immunity induction, macrophages invade the CNS within the first 12–18 h of pupal development [20]. For a summary of all genetic tools employed in this study see (S1 Fig). To test whether this coincides with the remodeling of the neural lamella, we analyzed the localization of its major components collagen IV, using a gene trap element, and laminin-γ, using specific antibodies, within the first 20 h of pupal development. Since panglial expression of *PGRP-LE* led to a pupal lethality in males, we only analyzed female brains. In control brains, the neural lamella was remodeled slowly during pupal development. When laminin-γ distribution was analyzed, almost no gaps were detected until 16 h after puparium formation (APF), while at 20 h APF, reduced laminin-γ staining intensity and distribution in neural lamella is clearly visible (Fig 1A). Upon panglial immunity induction loss of the laminin-γ signal at the neural lamella was already apparent at 8 h APF and overall, the neural lamella appeared thinner (Fig 1B). By contrast, when we determined the distribution of collagen IV, we noted loss of collagen IV:GFP expression in the neural lamella even earlier, already at 8 h APF in control pupae (Fig 1C). Upon immunity induction, the loss of the collagen IV

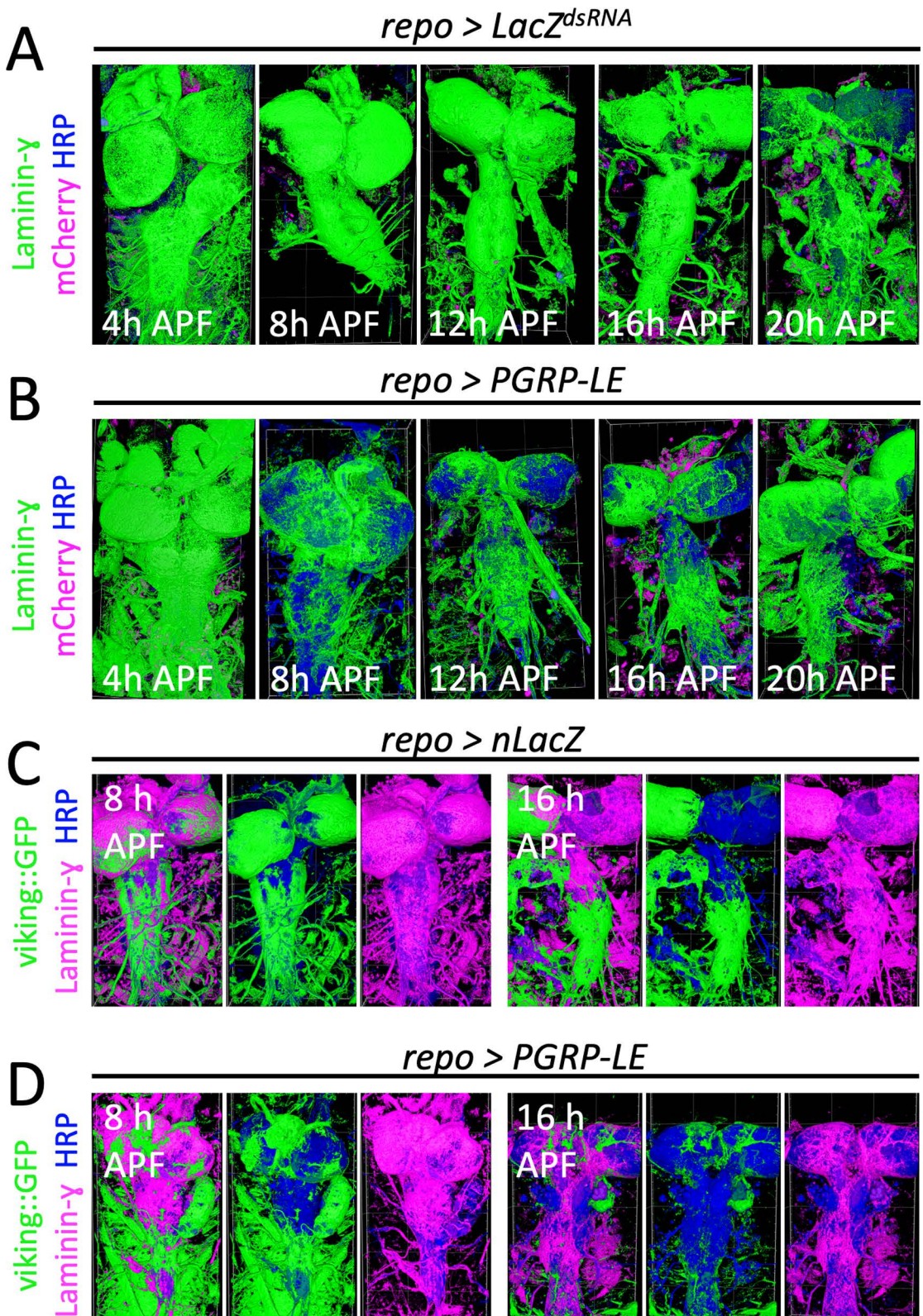

**Fig 1. Neural lamella is remodeled earlier upon immunity induction.** (**A, B**) Pupal filet preparations of different time points after pupariation formation (APF) as indicated stained for the neural lamella with Laminin-γ (green), horse radish

peroxidase (HRP) (blue) to label neuronal membranes and *mCherry* expression (magenta) directed by the macrophage marker *srpHemo-moe::3xmCherry*. To visualize the remodeling of the neural lamella, surfaces were generated using the arivis4D software. (A) Control animals expressing control dsRNA against *LacZ* in all glial cells. Note the onset of the neural lamella remodeling at 20 h APF. The following numbers were analyzed: $n_{4hAPF} = 5$, $n_{8hAPF} = 9$, $n_{12hAPF} = 7$, $n_{16hAPF} = 10$, $n_{20hAPF} = 6$. (B) Animals expressing Peptidoglycan recognition protein LE (*PGRP-LE)* in glial cells. The remodeling of the neural lamella starts at 8 h APF. The following numbers were analyzed: $n_{4hAPF} = 8$, $n_{8hAPF} = 7$, $n_{12hAPF} = 5$, $n_{16hAPF} = 4$, $n_{20hAPF} = 4$. (C) Pupal filet preparations of 8 and 16 h APF control animals stained for Collagen IV::GFP generated by the endogenous *viking* locus tagged with a green fluorescent protein (GFP) exon (green), Laminin-γ (magenta), and HRP (blue) to label neuronal membranes. To visualize the neural lamella, surfaces were generated using the arivis4D software. In controls, neural lamella remodeling starts at 8 h APF. Remodeling is more advanced at 16 h APF. *n* = 4. (D) Upon panglial immunity induction, the neural lamella remodeling starts at 8 h APF and is more advanced compared to controls. Note that the degradation of Collagen IV::GFP is generally more advanced compared to Laminin-γ. At 16 h APF the Collagen IV::GFP largely disappeared. *n* > 3.

fluorescence signal was more pronounced and was almost gone at 16 h APF (Fig 1D). Surprisingly, we consistently found that the presence of collagen IV::GFP enhanced the degradation of laminin-γ and, moreover, that collagen IV::GFP is less stable compared to laminin-γ (Fig 1). Therefore, we excluded the use of collagen IV::GFP and only determined the localization of laminin-γ as proxy for the neural lamella remodeling in all subsequent experiments. The reduction in distribution and signal intensity of two integral BM components, laminin-γ and collagen IV, using both an immunofluorescence staining and a genetic approach, respectively, suggests loss of the BM structural integrity, potentially through proteolytic degradation.

## Mmp1 and Mmp2 are upregulated in response to a panglial immunity induction

Next, we wondered how panglial immunity induction could trigger precocious neural lamella remodeling. Different proteins constituting the ECM have been shown to be cleaved by *Drosophila* MMPs. In *Drosophila*, only two MMPs are known, Mmp1 and Mmp2 [42].

To investigate whether MMP expression is upregulated during panglial immunity induction, we performed quantitative real-time PCR (qRT-PCR) on mRNA extracted from brains of the indicated age (Fig 2A and 2B and S1 Data). Both, *Mmp1* and *Mmp2* expression were analyzed during three different developmental stages (Fig 2A and 2B). Analysis of the relative expression levels demonstrated that upon panglial immunity induction *Mmp1* expression was upregulated during wandering third instar larval (wL3) stages as well as 8 h APF (Fig 2A). *Mmp2* expression, in contrast, was not upregulated in larval stages, but was significantly increased in early pupal stages. In later pupal stages, *Mmp2* expression level resumed back to control levels (Fig 2B).

We next tested where expression of *Drosophila* MMPs is altered in response to a panglial immunity induction using specific antibodies raised against *Drosophila* Mmp1 as well as a GFP-labeled Mmp2 construct. In control animals weak Mmp1 localization was detected at the BBB of the CNS of wL3, whereas no Mmp1 is found at the pupal BBB (Fig 2C, 2E and 2G). By contrast, upon immunity induction increased staining for Mmp1 was detected in wL3 brains as well as during 8 h APF old pupal brains (Fig 2D and 2F, arrowheads). Here, Mmp1 staining was mainly detected at the BBB (Fig 2D′ and 2K arrowheads). Correlating with the expression of Mmp1 we noted a change in CNS morphology similar what has been described for Mmp2 expression [44]. No change in Mmp1 staining was observed at later pupal stages when comparing control animals with panglial immunity induction animals (Fig 2G and 2H). Mmp2 was barely detectable in control young pupal brains (Fig 2I). In contrast, upon immunity induction Mmp2 can be detected at the BBB (Fig 2J and 2L, arrowheads). Moreover, we observed weak Mmp2 staining at the neural lamella (Fig 2L, asterisk).

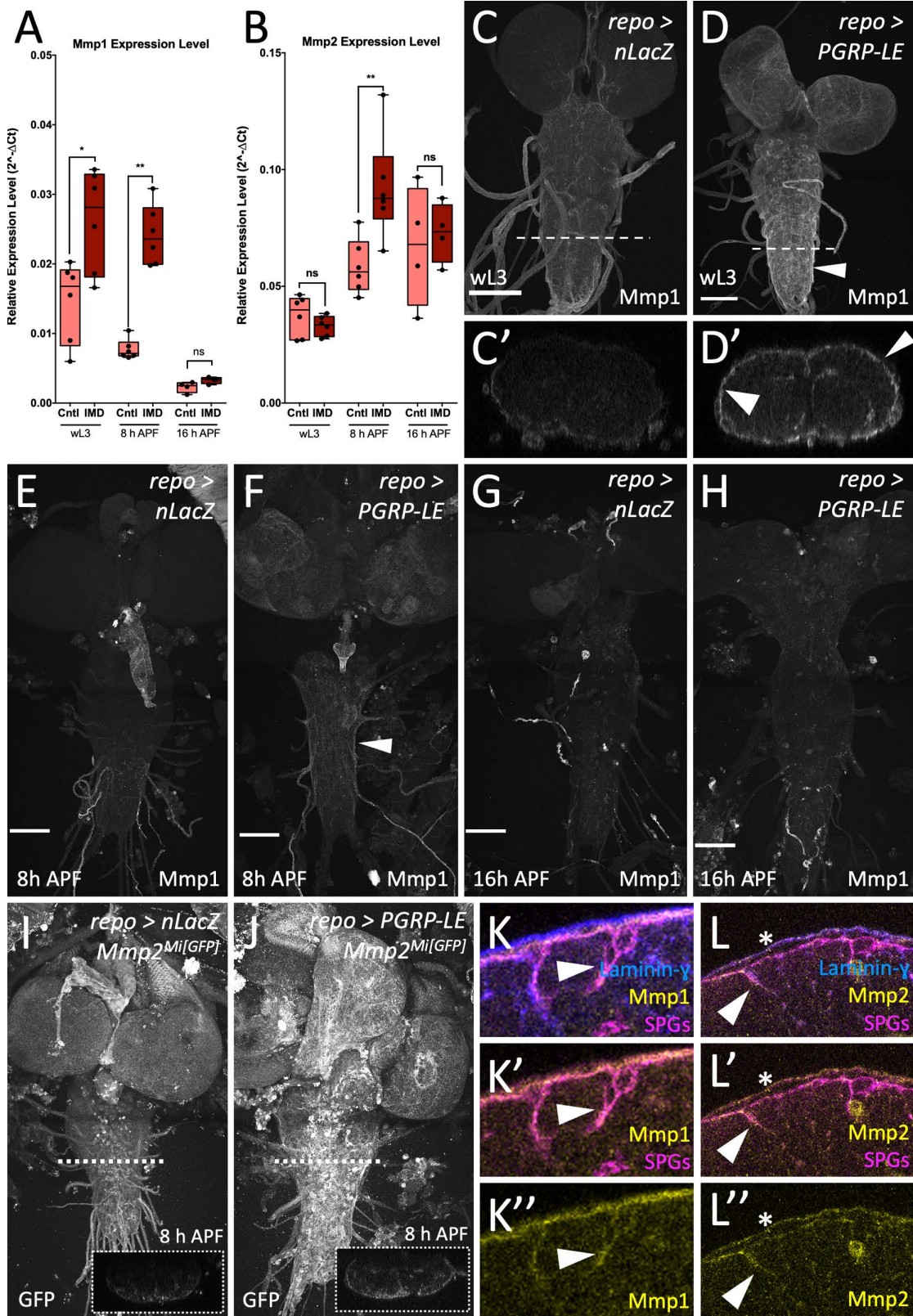

**Fig 2. MMPs are upregulated upon panglial immunity induction.** (**A, B**) Brains of different developmental stages of control and panglial immunity induction were submitted to quantitative real-time PCR (qRT-PCR) and tested for *Mmp1* and *Mmp2*

expression levels. Panglial expressed *nuclear LacZ* (*nLacZ*) was used as a control. *n* = 4–6 for all samples. wL3: wandering third instar larvae. h APF: hours after puparium formation. (A) *Mmp1* expression upon panglial immunity induction during wL3, 8 h and 16 h APF stages compared to controls. Mann–Whitney for all samples: wL3: $*P_{(Cntl\ vs.\ IMD)}$ = 0.0411; 8 h APF: $**P_{(Cntl\ vs.\ IMD)}$ = 0.0022; 16 h APF: $P_{(Cntl\ vs.\ IMD)}$ = 0.1143. (B) Expression of *Mmp2* in wL3 stages does not differ between controls and panglial immunity induction. At 8 h APF, *Mmp2* expression levels are upregulated upon panglial immunity induction, while at 16 h APF no difference to controls is detected. Mann–Whitney for all samples: wL3: $P_{(Cntl\ vs.\ IMD)}$ = 0.4740; 8 h APF: $**P_{(Cntl\ vs.\ IMD)}$ = 0.0087; 16 h APF: $P_{(Cntl\ vs.\ IMD)}$ > 0.9999. Ct values and quantification are shown in S1 Data. (C–H) Mmp1 localization in larval and pupal brains of the indicated age. (C–D′) Wandering third instar larval brains. Note the increased signal of Mmp1 at the surface of the CNS upon panglial immunity induction (D′, arrowheads). (E, F) At 8 h APF, Mmp1 staining is increased upon panglial immunity induction (F) compared to control brains (E). (G, H) At later pupal stages, Mmp1 staining does not differ between control (G) and panglial immunity induction (H). *n* > 5 brains were analyzed for all time points. Scale bars, 100 μm. (I, J) Mmp2 localization in 8 h APF pupal brains. (I) Control, (J) upon panglial immunity induction. The dotted box shows an orthogonal section taken at the position indicated. Note the increased Mmp2 signal at the BBB upon panglial immunity induction. Control *n* = 2, Immunity Induction *n* = 7. (K) Close-up of a wandering third instar larval brain during immunity induction with concomitant expression of *mdr65-tdTomato* (magenta) to label subperineurial glial cells (SPGs). The neural lamella was stained using Laminin-γ antibodies (blue). Localization of Mmp1 is shown in yellow. Note the co-localization of SPGs and the Mmp1 signal (arrowhead). (L) Close-up of an 8 h APF pupal brain during immunity induction with concomitant expression of *mdr65-tdTomato* (magenta) to label SPGs. The neural lamella was stained using Laminin-γ antibodies (blue). Localization of Mmp2 is visualized by endogenously GFP-tagged Mmp2. Mmp2 is detected at the SPGs (arrowhead), and the neural lamella (asterisk).

## Mmp1 and Mmp2 are required during macrophage invasion into the brain

To test the significance of increased *MMP* expression during larval and early pupal stages for macrophage invasion upon immunity induction, we first blocked the function of both *Drosophila* MMPs. For this, we employed upstream activating sequence (UAS)-driven *Drosophila Tissue inhibitor of metalloproteases* (*TIMP*) expression which efficiently interferes with the proteolytic activity of both MMPs [42]. In order to control for the presence of an additional UAS-element and thus for a possible reduction in the amount of PGRP-LE, we included the expression of an UAS-based *GFP^dsRNA* as control. When *TIMP* was co-expressed together with *PGRP-LE* to induce a panglial immunity response the number of invading macrophages was greatly decreased (Fig 3A and S2 Data). This suggests that ECM remodeling by MMPs occurs during macrophage invasion. In a next step, we performed a panglial knockdown of *Mmp1* or *Mmp2* during a panglial immunity induction. Interestingly, suppression of *Mmp1* led to a 90% reduction in the number of invading macrophages (Fig 3B and S2 Data). Quite similar, suppression of *Mmp2* expression resulted in a highly significant reduction in the number of invaded macrophages (Fig 3B). In conclusion, Mmp1 and Mmp2 are both required for macrophage invasion into the *Drosophila* brain suggesting that the remodeling of the neural lamella is needed to allow invasive migration. To directly show whether MMP expression influences the stability of the neural lamella, we stained for laminin-γ localization following *Mmp1* or *Mmp2* knockdown and found a slightly delayed neural lamella remodeling in 8 h APF pupae in both knockdown paradigms (Fig 3C).

## Regulation of MMP expression

In vertebrates, expression of *MMPs* can be regulated by a number of signaling pathways including JNK [14,47,48]. For *Drosophila*, we demonstrated that the JNK pathway is not involved in macrophage invasion upon induction of panglial immunity [20]. In addition, we employed a JNK reporter (*puc::GFP*) to track JNK signaling activation in the brain of wandering third instar control larvae and those undergoing panglial immunity induction. No significant differences were observed between controls and a panglial immunity induction (S2A–S2C Fig and S8 Data). This was further corroborated by analyzing Mmp1 localization upon panglial immunity induction with a concomitant knockdown of *basket*, an essential component of the JNK signaling pathway (S2D and S2E Fig). Since Mmp1 localization was

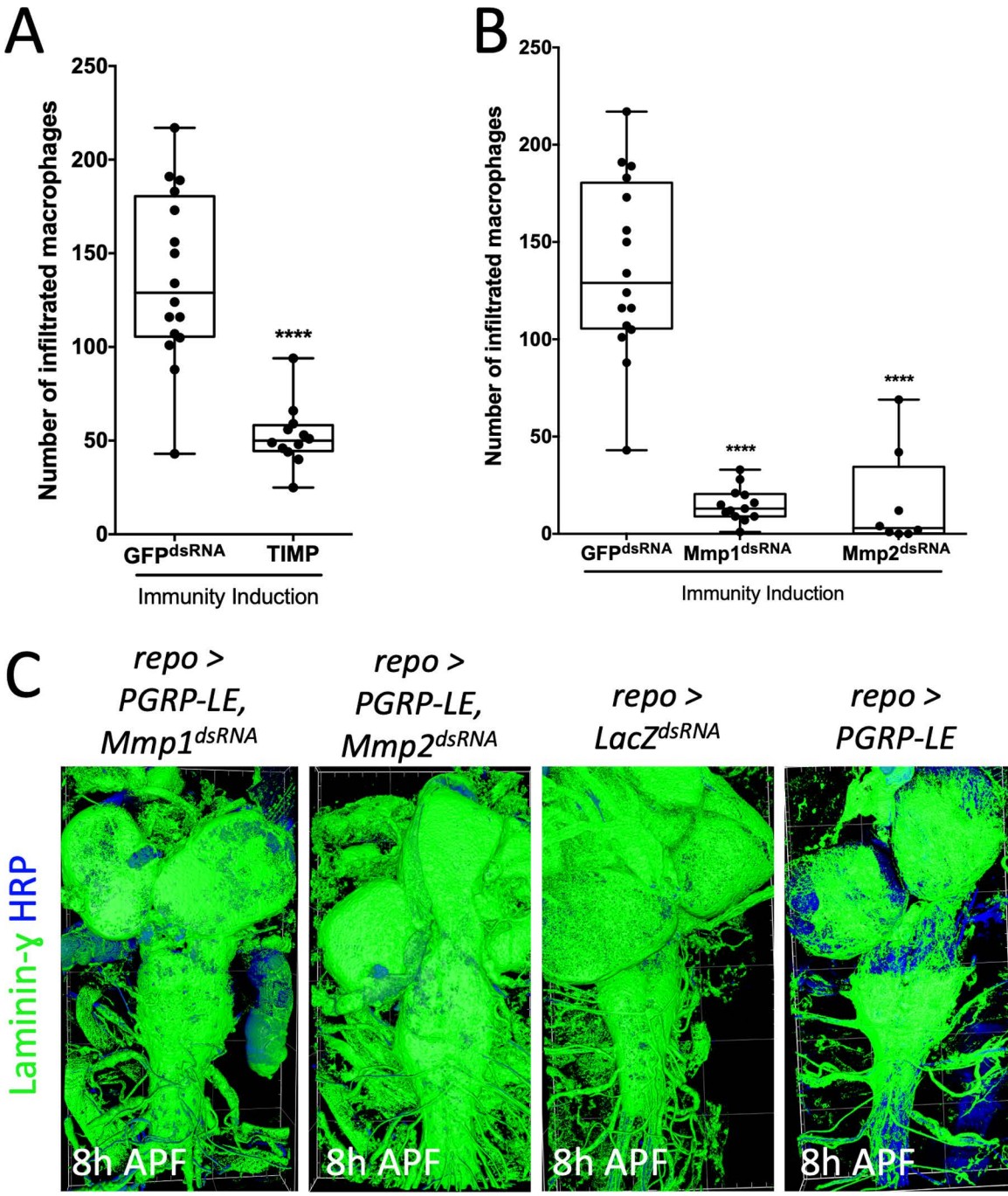

**Fig 3. Activity of matrix metalloproteinases (MMPs) is needed for sufficient macrophage invasion.** (**A**) Number of infiltrated macrophages within the brain upon panglial immunity induction of flies co-expressing either the *Tissue inhibitor of metalloproteases* (*TIMP*) or a double-stranded RNA against *GFP*. For counting, the macrophage marker *srpHemo-H2A::3xmCherry* was used and data were analyzed using the arivis4D software. Unpaired *t* test, ****$P < 0.0001$. *GFP*dsRNA $n = 16$, *TIMP* $n = 12$. (**B**) Quantification of macrophage invasion upon panglial immunity induction with a dsRNA directed against *GFP* as control and a dsRNA directed against *Mmp1* or *Mmp2* in all glial cells. Kruskal–Wallis test followed by Dunn's multiple comparison test was performed for all samples generating adjusted *P*-values: ****$P_{(GFP\,dsRNA\,vs.\,Mmp1\,dsRNA)} < 0.0001$, ****$P_{(GFP\,dsRNA\,vs.\,Mmp2\,dsRNA)} < 0.0001$. Control $n = 16$, Mmp1dsRNA $n = 13$, Mmp2dsRNA $n = 8$. For all counting data, see S2 Data. (**C**) CNS filet preparations of genotypes indicated stained for Laminin-γ (green) and HRP (blue, showing neuronal membranes) at 8 h APF. For visualization of the neural lamella, surfaces were generated with arivis4D of the Laminin-γ staining. The following numbers were analyzed: $n_{Mmp1KD} = 6$, $n_{Mmp2KD} = 4$, $n_{LacZKD} = 9$, $n_{Imd+GFPKD} = 4$. Note, the rather intact neural lamella upon *Mmp1*dsRNA or *Mmp2*dsRNA expression.

not altered when *basket* expression was silenced, we conclude that JNK signaling is not involved in regulating *Mmp1* expression. Therefore, alternative signaling pathways are likely implicated.

## JAK/STAT signaling is activated in BBB glial cells upon immunity induction

In vertebrates, expression of *MMPs* can also be regulated by JAK/STAT signaling [14,47,48]. In addition, tumor cell-derived cytokines can activate JAK/STAT signaling in the *Drosophila* BBB which in turn triggers the opening of the BBB [49]. We therefore tested whether JAK/STAT signaling is also involved in the macrophage invasion upon a panglial immunity induction.

First, we determined if JAK/STAT signaling is activated upon immunity induction. For this, we employed an established transgenic STAT92E signaling activity reporter [50]. Upon immunity induction an almost 3-fold increase of STAT92E activity was detected in cells of the BBB and neuropil-associated glia of third instar larvae, resembling Mmp1 localization (Fig 4A–4C and S3 Data). This increase in STAT92E activity in the brain remained in 8- and 16-h old pupae (S3A–S3D Fig). In 24-h old pupae, no more STAT92E activity is detected in the nervous system. The BBB is comprised of PGs and SPGs. To determine which of the two cell types forming the *Drosophila* BBB activate STAT92E signaling, we included a *mdr65-tdTomato* construct with highly specific expression in SPGs [51]. Co-labeling experiments demonstrated expression of the STAT92E reporter in both glial cell types (Fig 4D). STAT92E::GFP is activated in SPGs which projects fine processes into the CNS cortex as well into the PG layer (Fig 4D′–4D‴, arrowhead). The PGs also activate the STAT92E reporter (Fig 4D′–4D‴, arrow). Thus, both cell types of the BBB can activate JAK/STAT signaling.

## JAK/STAT signaling is required during macrophage invasion

Given the close spatial association of JAK/STAT signaling and the neural lamella, we tested if JAK/STAT signaling affects neural lamella organization during immunity induction. The key effector of JAK/STAT signaling is the transcription factor STAT92E. Panglial knockdown of *STAT92E* during immunity induction indeed stabilized the neural lamella during pupal stages (Fig 5A and 5B). Moreover, panglial *STAT92E* knockdown in wild type animals shifted the onset of ECM remodeling to older pupal stages compared to controls (S3E and S3F Fig).

Having demonstrated that panglial immunity induction triggers STAT92E activity, we hypothesized that JAK/STAT signaling is needed during macrophage invasion. JAK/STAT signaling is activated by binding of the interleukin-like Upd ligands to the Dome receptor. This leads to recruitment of the Hop kinase, and subsequent phosphorylation and dimerization of STAT92E. To directly test the involvement of this signaling cascade for macrophage invasion, we suppressed the expression of *STAT92E* during a panglial immunity induction. This treatment efficiently blocked macrophage invasion, and only few macrophages were found within the CNS (STAT92E$_{mean}$ = 6.3 macrophages/CNS, $n$ = 8; Control$_{mean}$ = 137 macrophages/CNS, $n$ = 16) (Fig 5C and S4 Data). To test whether knockdown of STAT92E possibly reduces the overall number of macrophages in the hemolymph, we performed a dot blot assaying total expression levels of the same macrophage marker protein that was used for counting invading macrophages (Fig 5D and S4 Data). This analysis indicated an increase in the total number of macrophages, rather than a reduction. We also tested whether knockdown of either *dome* or *hop* in glial cells during a panglial immunity induction caused a similarly

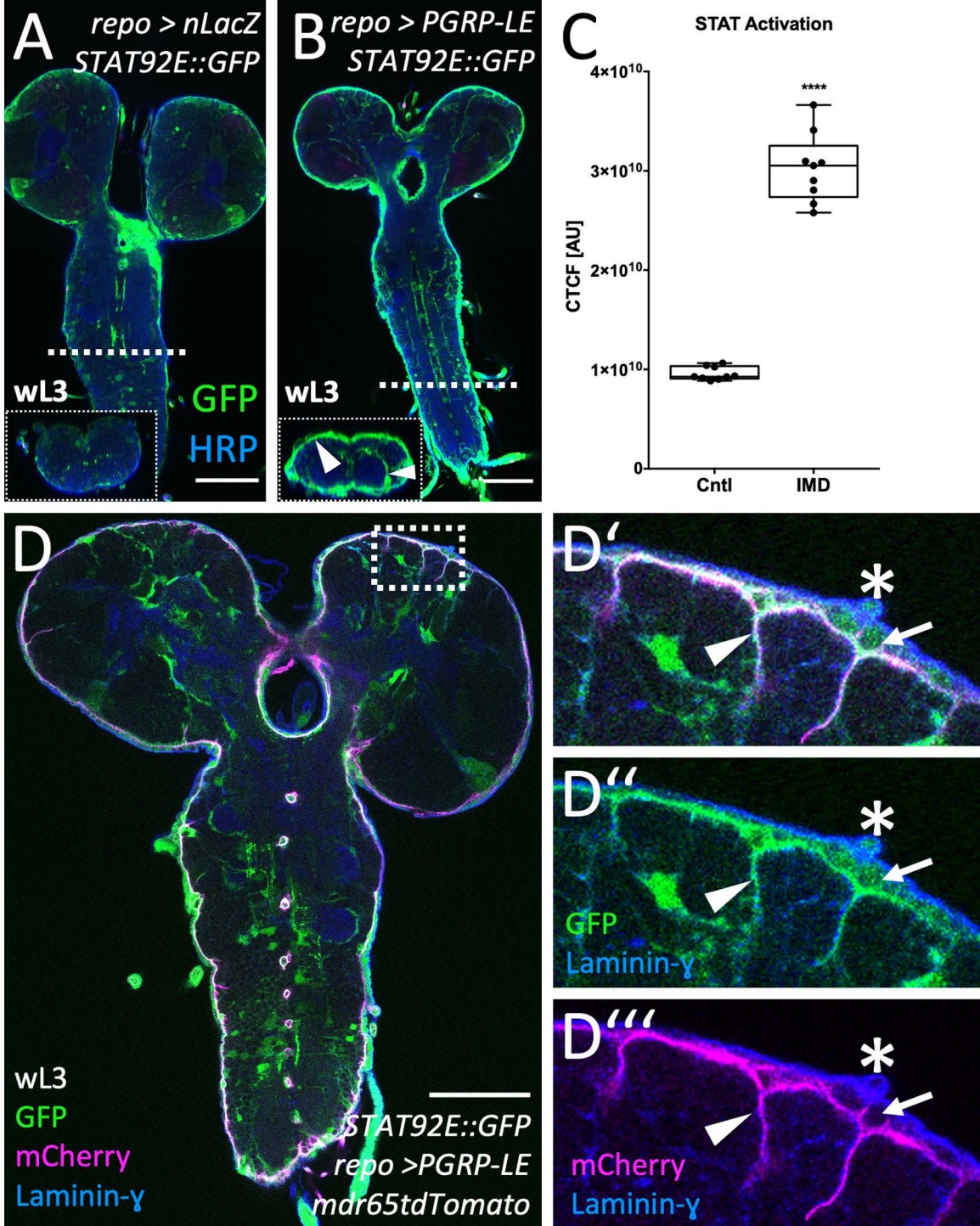

**Fig 4. Janus kinase/signal transducer and activator of transcription (JAK/STAT) signaling is activated in BBB glial cells upon immunity induction.** (**A**, **B**) Third instar larval brains expressing a JAK/STAT signaling reporter with and without panglial immunity induction. JAK/STAT signaling activation is shown in green, neuronal membranes are in blue (anti-HRP staining). Scale bars, 100 μm. (**C**) Corresponding measurements of JAK/STAT activation. The corrected total cell fluorescence (CTCF) was calculated in arbitrary units (AU), see Materials and methods for details. Mann–Whitney, ****$P_{(Control\ vs.\ IMD)} < 0.0001$. $n = 9$. For raw data and quantification see S3 Data. (**D**–**D‴**) Simultaneous expression of the JAK/STAT signaling reporter (green) and a subperineurial glial cell (SPG) marker (*mdr65-tdTomato* stained with dsRed (magenta)) following panglial immunity induction. The neural lamella is labelled using anti-Laminin-γ antibodies (blue). STAT92E activity is detected in SPGs (D′–D‴, arrowheads), and in perineurial glial cells (arrow), $n > 5$. The asterisk denotes a trachea which also has STAT92E activity. Scale bar, 100 μm.

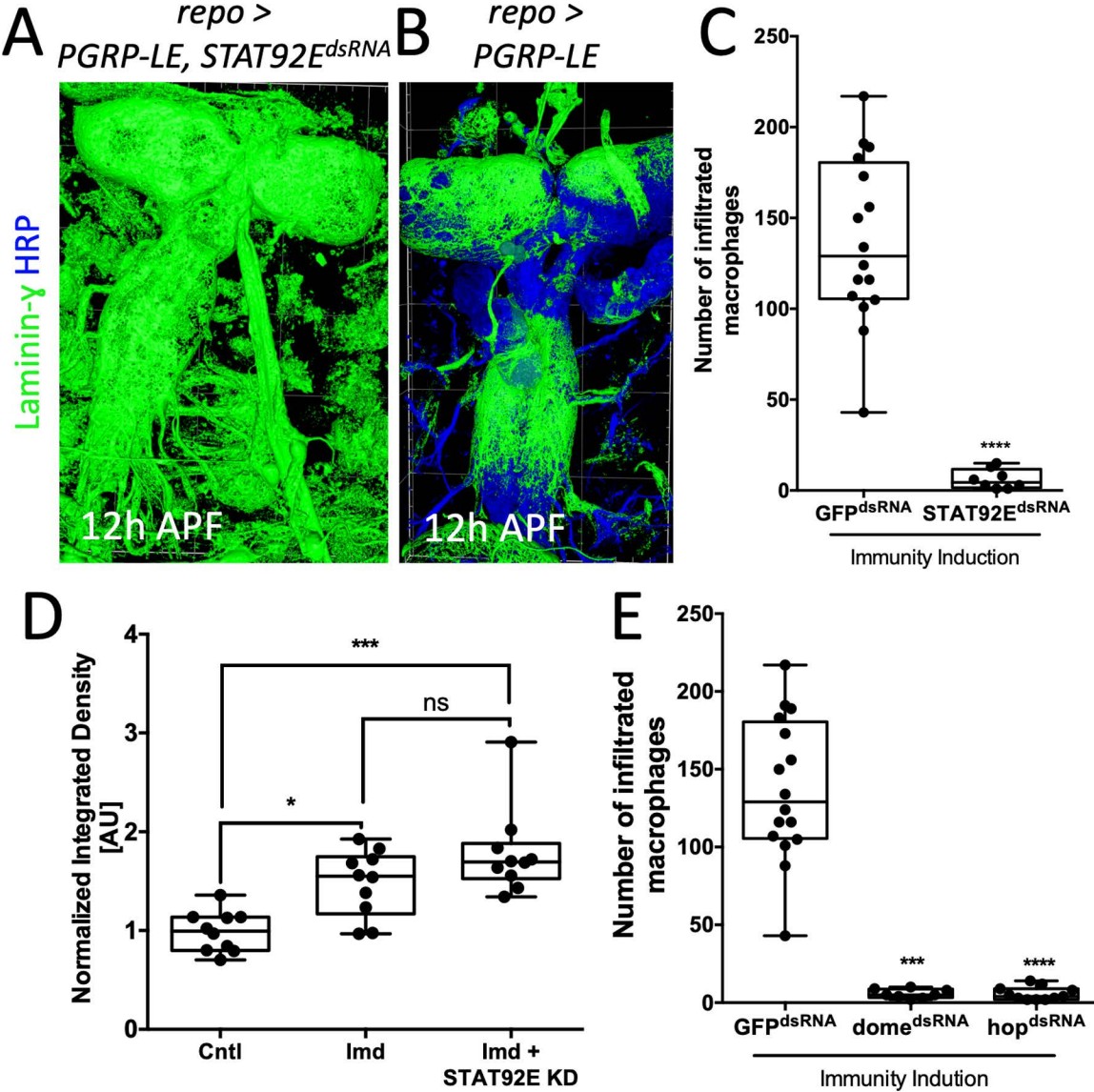

**Fig 5. JAK/STAT signaling is needed for a sufficient macrophage invasion.** (**A**, **B**) CNS filet preparations of 12 h APF pupae of the genotypes indicated stained for Laminin-γ (green, neural lamella) and HRP (blue, neuronal membranes). The neural lamella appears intact upon a panglial *STAT92E* downregulation. Laminin-γ surfaces were generated with arivis4D. (**C**) Downregulation of *STAT92E* in glial cells during a panglial immunity induction decreases the number of infiltrated macrophages. Mann–Whitney, ****$P < 0.0001$. *GFP*dsRNA $n = 16$, *STAT92E*dsRNA $n = 8$. For counting data in C and E, see S4 Data. (**D**) Quantification of dot blots of controls expressing panglial *nLacZ*, panglial immunity induction (Imd) or panglial immunity induction with concomitant expression of a *STAT92E*dsRNA. All genotypes additionally express *srpHemo-H2A::3xmCherry* to label all macrophages. To quantify total macrophage numbers, protein isolations of whole single larvae were stained using anti-dsRed and anti-Tubulin. Every data point is the average of three technical replicates normalized to tubulin. Kruskal–Wallis test followed by Dunn's multiple comparison test was performed for all samples generating adjusted *P*-values: *$P_{(Cntl\ vs.\ Imd)} = 0.0132$, $P_{(Imd\ vs.\ Imd\ +\ STAT92E\ KD)} = 0.4652$, ***$P_{(Cntl\ vs.\ Imd\ +\ STAT92E\ KD)} = 0.0002$. $n = 10$ for each genotype. For quantification see S4 Data, raw images are in S1 Raw Images. (**E**) Downregulation of *domeless* (*dome*) or *hopscotch* (*hop*) in glial cells during a panglial immunity induction decreases the number of invaded macrophages. Kruskal–Wallis test followed by Dunn's multiple comparison test was performed for all samples generating adjusted *P*-values: ***$P_{(dome\ dsRNA\ vs.\ GFP\ dsRNA)} < 0.0003$, ****$P_{(hop\ dsRNA\ vs.\ GFP\ dsRNA)} < 0.0001$. *GFP*dsRNA $n = 16$, *dome*dsRNA $n = 8$, *hop*dsRNA $n = 11$.

strong decrease of the macrophage invasion rate and found similar effects (dome$_{mean}$ = 6 macrophages/CNS, $n$ = 8; hop$_{mean}$ = 6 macrophages/CNS, $n$ = 11) (Fig 5E and S4 Data).

## JAK/STAT signaling contributes to regulation of MMP expression

As JAK/STAT signaling was shown to play an essential role for macrophage invasion and neural lamella remodeling, we wanted to determine whether it might regulate *MMP* expression. While a panglial immunity induction caused an increase of Mmp1 as well as Mmp2 localization at the BBB (see Fig 2), a strong decrease of Mmp1 protein signal was observed upon silencing of *STAT92E* by RNAi (Fig 6A and 6B). To further verify if STAT92E functions upstream of induced *MMP* expression, we investigated the expression level of *Mmp1* and *Mmp2* upon panglial immunity induction with a concomitant *STAT92E* knockdown in larval

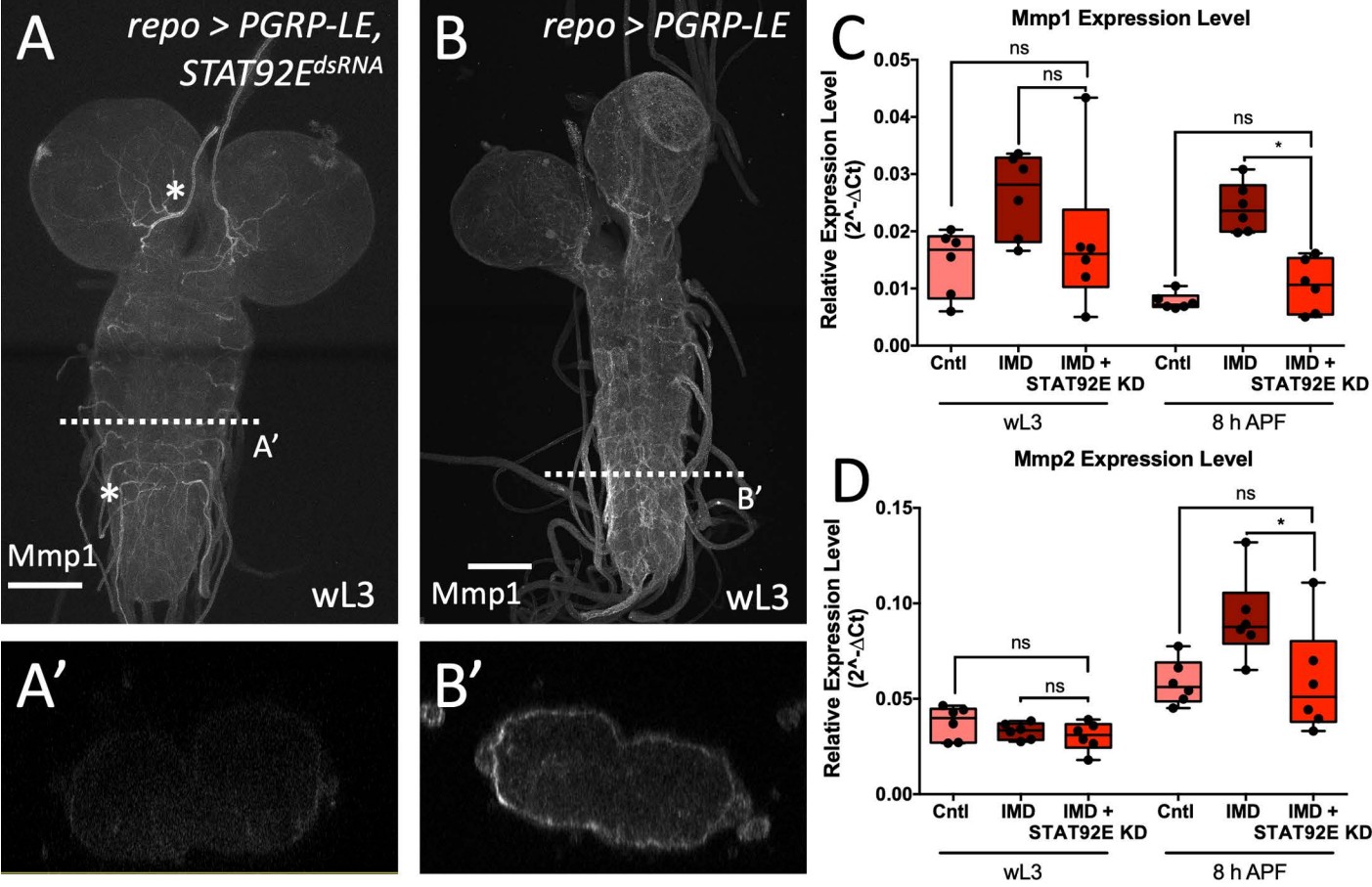

**Fig 6. JAK/STAT signaling can activate MMP expression.** (**A**, **B**) Larval CNS with a panglial immunity induction (B) and a simultaneous *STAT92E* knockdown (A) stained for Mmp1 (white). Upon *STAT92E* knockdown, only very low Mmp1 staining is detected mainly localized to the trachea (asterisks). $n$ > 5. (**C**, **D**) The CNS of larvae and young pupae (8 h APF) were analyzed for the expression level of *Mmp1* (C) and *Mmp2* (D) in controls (panglial *nLacZ* expression), panglial immunity induction and panglial immunity induction with a simultaneous *STAT92E* knockdown. $n$ = 6 for all samples. For raw data and quantification see S5 Data. (C) *Mmp1* expression levels significantly decrease during *STAT92E* downregulation compared to the sole panglial immunity induction in pupal stages. Kruskal–Wallis test followed by Dunn's multiple comparison test was performed for all samples generating adjusted *P*-values: wL3: $P_{(Cntl \, vs. \, IMD \, STAT92E \, KD)}$ > 0.9999, $P_{(IMD \, vs. \, IMD \, STAT92E \, KD)}$ = 0.1169; 8 h APF: $P_{(Cntl \, vs. \, IMD \, STAT92E \, KD)}$ > 0.9999, *$P_{(IMD \, vs. \, IMD \, STAT92E \, KD)}$ = 0.0162. (D) No significant difference in *Mmp2* expression level was detected in larval stages. In pupal stages, *Mmp2* expression decreases significantly upon *STAT92E* downregulation and reaches control levels. Kruskal–Wallis test followed by Dunn's multiple comparison test was performed for all samples generating adjusted *P*-values: wL3: $P_{(Cntl \, vs. \, IMD \, STAT92E \, KD)}$ = 0.3034, $P_{(IMD \, vs. \, IMD \, STAT92E \, KD)}$ > 0.9999; 8 h APF: $P_{(Cntl \, vs. \, IMD \, STAT92E \, KD)}$ > 0.9999, *$P_{(IMD \, vs. \, IMD \, STAT92E \, KD)}$ = 0.0348.

and early pupal stages. Downregulation of *STAT92E* reduced *Mmp1* expression levels at larval and pupal stages; however, only significantly in pupal stages which did not differ significantly to controls (Fig 6C and S5 Data). By contrast, *Mmp2* expression level did not change in larval stages upon a panglial *STAT92E* knockdown. However, in early pupal stages *Mmp2* expression levels were significantly reduced compared to the panglial immunity induction and did not vary significantly from control levels (Fig 6D and S5 Data). In conclusion, both Mmp1 protein expression as well as *Mmp2* mRNA expression can be influenced by JAK/STAT signaling.

## PGRP-LE expression affects expression of Unpaired ligands

The JAK/STAT signaling receptor Dome is activated by binding of interleukin-like signaling proteins encoded by the *upd* genes *upd1*, *upd2* and *upd3* [38]. To define the specific *upd* gene responsible for STAT92E activation in the BBB following immunity induction, we determined the expression level of *upd1*, *upd2* and *upd3* in the CNS of control animals and upon immunity induction (Fig 7A and S6 Data). In all stages analyzed, *upd1* expression was slightly reduced by immunity induction. By contrast, *upd2* and *upd3* expression was upregulated compared to controls, which was most prominent in wandering third instar larvae (Fig 7A).

Having shown that JAK/STAT is activated following immunity induction and *upd* expression is increased, we tested how the different unpaired cytokines modulate macrophage invasion during panglial immunity induction. *upd1* mutants are homozygous lethal and thus could not be analyzed, while *upd2* and *upd3* mutants are homozygous viable [39,52]. Both single mutants showed an intermediate decrease in the number of macrophages invading the pupal CNS during panglial immunity induction, suggesting that both genes are required for invasion of macrophages in a non-redundant manner. According, a *upd2*, *upd3* double mutant resulted in the strongest decrease in macrophage invasion (Fig 7B and S6 Data).

## ECM remodeling is needed for sufficient macrophage invasion

We previously showed that invasion of peripheral macrophages into the CNS during pupal stages not only depends on upregulation of the *PDGF/VEGF-related factor Pvf2*, but, moreover, ectopic expression of *Pvf2* within CNS cells is sufficient to trigger the invasion of macrophages [20]. Interestingly, macrophages invaded the CNS slightly later during pupal development as compared to *PGRP-LE* expression (Fig 8A and S7 Data). We now tested whether panglial *Pvf2* expression also results in JAK/STAT activation. Staining for laminin-γ upon panglial *Pvf2* expression revealed normal neural lamella remodeling during pupal stages (S4A Fig), indicating no enhanced neural lamella remodeling. Also, no significant activation of JAK/STAT signaling was observed upon panglial Pvf2 expression neither in third instar larval (Fig 8B and S7 Data) nor pupal brains (S4B and S4C Fig). Unexpectedly, when we downregulated *STAT92E* during panglial *Pvf2* expression and analyzed the invasion of macrophages into the CNS, we noticed a ~ 10-fold reduction in the number of invaded macrophages compared to panglial *Pvf2* expression (Fig 8C and S7 Data). These results suggest that the remodeling of the neural lamella that is normally triggered by JAK/STAT signaling is required to enable early macrophage invasion into the CNS.

## Forced MMP expression in perineurial glia allows precocious macrophage invasion

To further analyze the importance of the ECM remodeling for the macrophage invasion, we expressed *Mmp1* or *Mmp2* in glial cells of the BBB. To avoid developmental consequences of early *MMP* expression [44], we restricted their expression to late larval stages by including a

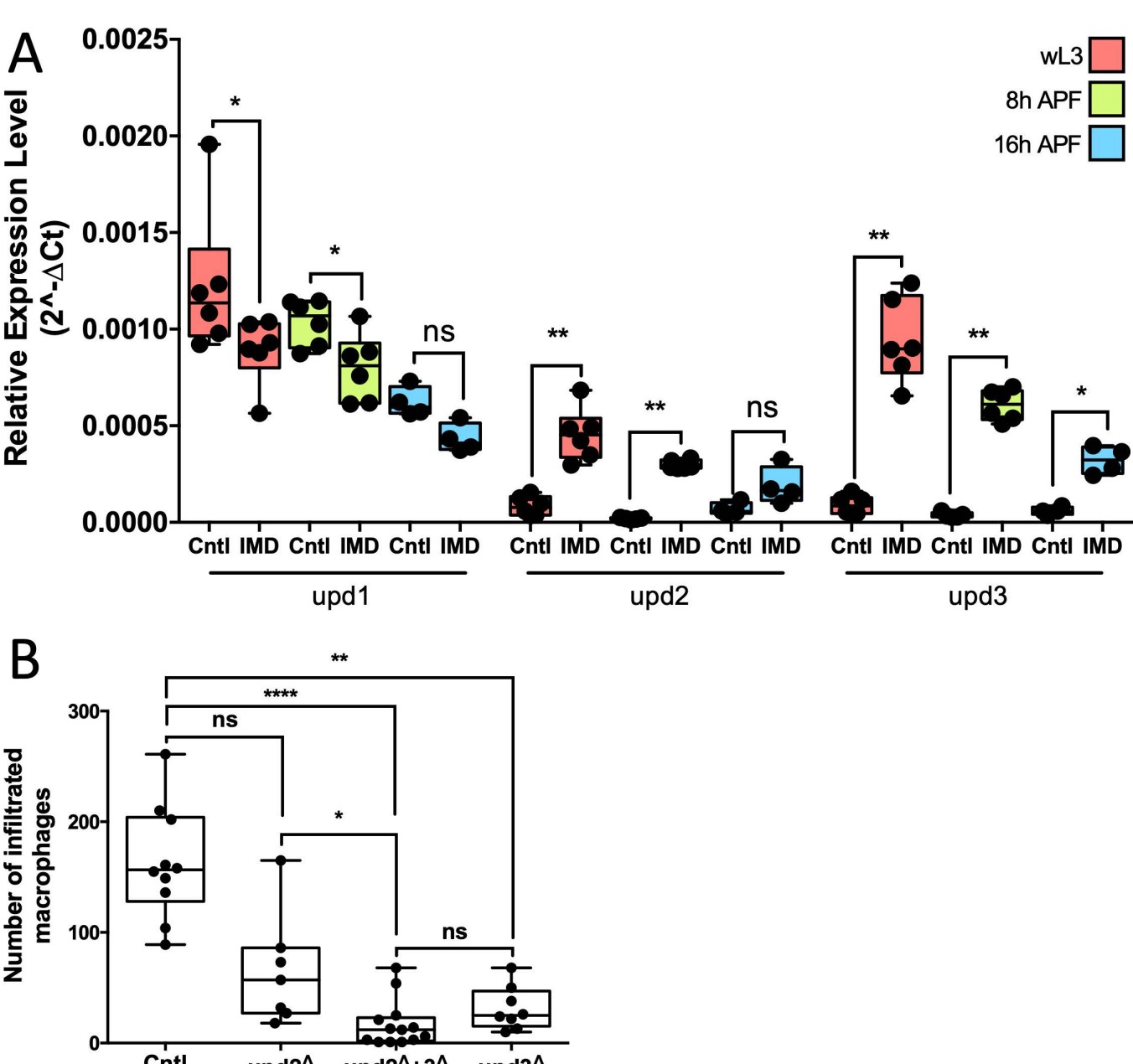

**Fig 7. Unpaired cytokines are upregulated upon immunity induction.** (**A**) Dissected CNS preparations of control and panglial immunity animals of indicated ages were analyzed for the expression of the *unpaired* (*upd*) cytokines using quantitative real-time PCR (qRT-PCR). Panglial expressed *nLacZ* was used as a control. While the expression of *upd1* is decreased in all stages, *upd2* and *upd3* are both upregulated during a panglial immunity induction in all stages. Mann–Whitney for all samples; *upd1*: *$P_{wL3}$ = 0.0411, *$P_{8h\,APF}$ = 0.0260, $P_{16h\,APF}$ = 0.1143; *upd2*: **$P_{wL3}$ = 0.0022, **$P_{8h\,APF}$ = 0.0043, $P_{16h\,APF}$ = 0.0571; *upd3*: **$P_{wL3}$ = 0.0022, **$P_{8h\,APF}$ = 0.0043, *$P_{16h\,APF}$ = 0.0286. $n$ = 4–6. For raw data and quantification see S6 Data. (**B**) Quantification of invasion rates during panglial immunity induction in different *upd* mutants. Single mutations in *upd2* or *upd3* decrease macrophage invasion. Double *upd2*, *upd3* mutants decrease the number of invaded macrophages even further. Kruskal–Wallis test followed by Dunn's multiple comparison test was performed for all samples generating adjusted *P*-values: $P_{(upd2\Delta\ vs.\ Cntl)}$ = 0.2082, ****$P_{(upd2\Delta+upd3\Delta\ vs.\ Cntl)}$ < 0.0001, **$P_{(upd3\Delta\ vs.\ Cntl)}$ = 0.0043, *$P_{(upd2\Delta\ vs.\ upd2\Delta+upd3\Delta)}$ = 0.0331, $P_{(upd3\Delta\ vs.\ upd2\Delta+upd3\Delta)}$ = 0.6029. Control $n$ = 10, $upd2\Delta$ $n$ = 7, $upd2\Delta + upd3\Delta$ $n$ = 13, $upd3\Delta$ $n$ = 8. For raw data see S6 Data.

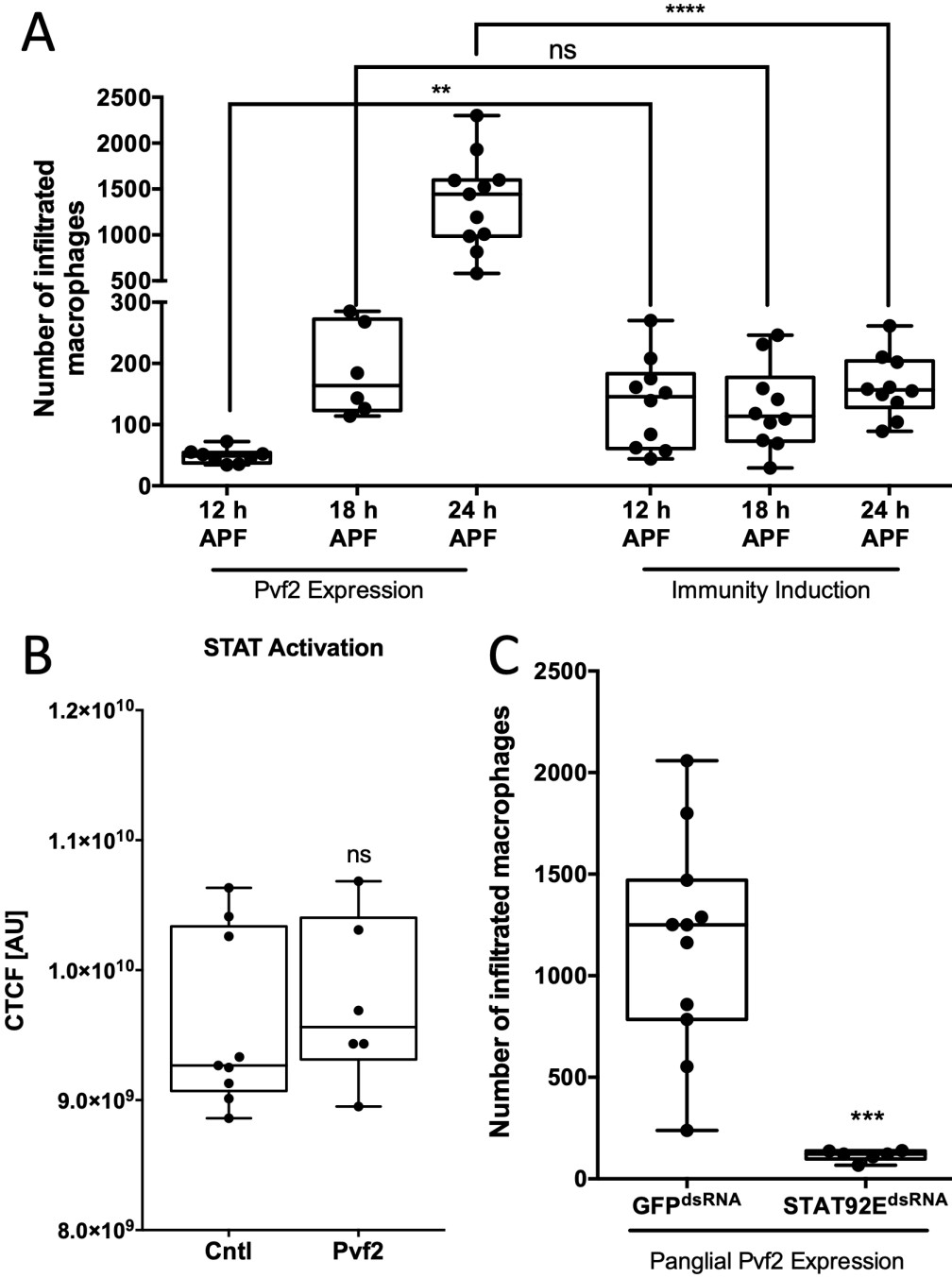

**Fig 8. Neural lamella remodeling is needed for macrophage invasion.** (**A**) Quantification of macrophage invasion during different developmental stages upon panglial *Pvf2* expression or panglial immunity induction. Upon a panglial *Pvf2* expression macrophages invade the central nervous system more abundantly in later pupal stages compared to a panglial immunity induction. Mann–Whitney, $**P_{12h\,APF} = 0.0021$, $P_{18h\,APF} = 0.0934$, $****P_{24h\,APF} < 0.0001$. Pvf2 12h APF $n = 8$, Pvf2 18h APF $n = 6$, Pvf2 24h APF $n = 11$, all samples of immunity induction $n = 10$. (**B**) Measurement of JAK/STAT activity in brains of wandering third instar larvae, with panglial expression of either *nLacZ* or *Pvf2*. No significant difference is detected. Mann–Whitney, $P_{(Cntl\,vs.\,Pvf2)} = 0.3864$. Pvf2 $n = 6$, control $n = 9$. (**C**) Panglial expression of *Pvf2* and concomitant knockdown of *STAT92E* decreases the number of invaded macrophages by ~ 10-fold. Mann–Whitney, $***P = 0.0002$. *GFP*dsRNA $n = 11$, *STAT92E*dsRNA $n = 6$. For all values and quantifications see S7 Data.

temperature sensitive Gal80 element [53]. To concomitantly trigger macrophage invasion, we established a Q-transcription factor/QF upstream activating sequences (QF/QUAS)-based *Pvf2* expression system [54] to enable expression independent of Gal4. Additionally, we included a *srpHemo-moe::3xmCherry* construct to label macrophages [55]. Panglial expression of *Pvf2* under QUAS resulted in a robust macrophage invasion during pupal stages (S4D Fig).

We then expressed *Pvf2* in all glial cells and simultaneously induced the expression of either *Mmp1* or *Mmp2* during L2/L3 stages in the different glial cells of the BBB. Expression of *Mmp1* specifically in SPGs using *moody-Gal4* [19] did not noticeably influence laminin-γ distribution in the larval CNS and did not result in macrophage invasion (*n* = 19) (S5A–S5A″ Fig, asterisk). Expression of *Mmp2* in SPGs disrupted neural lamella integrity during larval stages (compare S5A′-S5B″ Fig, asterisks). It did not allow breaching of the barrier and macrophages only adhered to the outer surface of the BBB (*n* = 5) (S5B–S5B″ Fig, arrowheads)

When we expressed *Mmp1* specifically in PGs using 2x*aptE01-Gal4*, the neural lamella appeared intact (Fig 9A and 9B). However, upon *Mmp2* expression in PGs, the neural lamella was partially degraded and the CNS was distorted as reported before [44] (Fig 9C and 9D). Interestingly, upon the expression of *MMPs* only in PGs, macrophages were able to invade a *Pvf2* expressing nervous system during larval stages. Ectopic expression of *Mmp1* led to a macrophage invasion in 28% of analyzed brains (*n* = 14) (Fig 9A and 9B), while expression of *Mmp2* resulted in a macrophage invasion in 58% of analyzed brains (*n* = 17) (Fig 9C and 9D).

Thus, our data support a model where immunity induction triggers JAK/STAT signaling to activate expression of MMPs that eventually remodel the ECM of the neural lamella (Fig 10). While this process also occurs in normal development, its precocious activation is sufficient to the allow invasion of few macrophages.

## Discussion

Bacterial infection of the *Drosophila* brain can occur in nature, but the number of invading macrophages is very low (1–2 macrophages, see [20]). Panglial expression of *PGRP-LE* triggers the expression of a battery of secreted proteins downstream of NF-κB and causes a pathological invasion of > 50 macrophages into the *Drosophila* brain. AMPs are involved in directly fighting pathogen invasion, while *Pvf2* expression efficiently recruits macrophages across the BBB into the CNS. Finally, as we show here, PGRP-LE also induces the expression of cytokines of the Upd family, which activate JAK/STAT signaling specifically in the BBB. This in turn activates the expression of MMPs that participate in remodeling of the dense ECM at the outer surface of the nervous system, which is needed for efficient macrophage invasion.

The role of JAK/STAT signaling in neuroinflammatory processes is not limited to *Drosophila* but is also found in the mammalian brain. Here, activated JAK/STAT signaling promotes neuroinflammation by initiating innate immunity, but also constrains neuroinflammatory responses. In neurons, activated JAK/STAT signaling was shown to be neuroprotective during Alzheimer's disease [56,57]. By contrast, JAK/STAT signaling can also promote cell death and contribute to brain damage. Recently, it was shown that STAT3 activation in reactive astrocytes promotes secretion of Serpina3n/α1ACT leading to BBB disruption by a decrease of the tight junction protein Claudin-5 and upregulation of the vascular cell adhesion protein 1 (VCAM-1) in BBB endothelial cells [15].

While aging is one major risk factor for many neurodegenerative diseases and is accompanied by a low-grade systemic inflammation [58,59], it does not appear to be relevant in the *Drosophila* system. In flies, macrophages preferentially invade the nervous system only during early pupal development. Why there is such a defined temporal window allowing macrophage invasion into the brain is currently not understood. Possibly, macrophages are primed

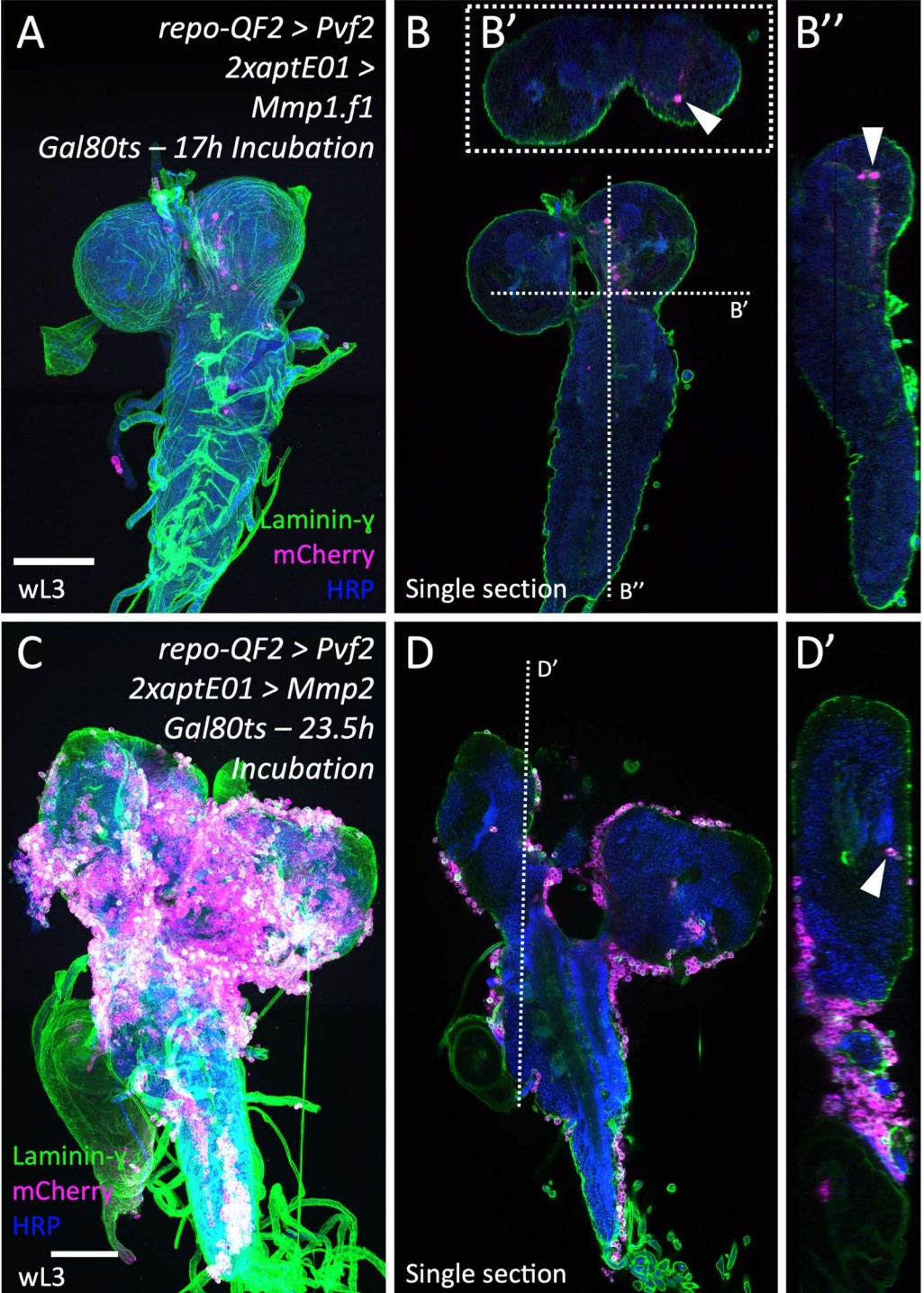

**Fig 9. MMP expression in perineurial glial enables precocious macrophage invasion.** Larval brains stained for Laminin-γ (green) to visualize the neural lamella, HRP (blue) to label neuronal membranes, and the macrophage marker mCherry (magenta). Scale bars, 100 μm. (**A–B″**) Expression of *Mmp1* directed by the perineurial driver *2xaptE01-Gal4* together with panglial *Pvf2* expression leads to a macrophage invasion into the CNS during larval stages. *Mmp1* expression was restricted to late larval stages by a ubiquitously expressed *Gal80*ts and an appropriate temperature regime. Note the invaded macrophages in the larval CNS (**B–B″**, arrowheads). *n* = 14. (**C–D′**) Induced expression of *Mmp2* in perineurial glial cells restricted by a *tub-P-Gal80*ts to late larval stages with a simultaneous panglial *Pvf2* expression attracts many macrophages to the degrading neural lamella, and some invade the CNS (**C–D′**, arrowhead). The VNC and lobes appear distorted. *n* = 17.

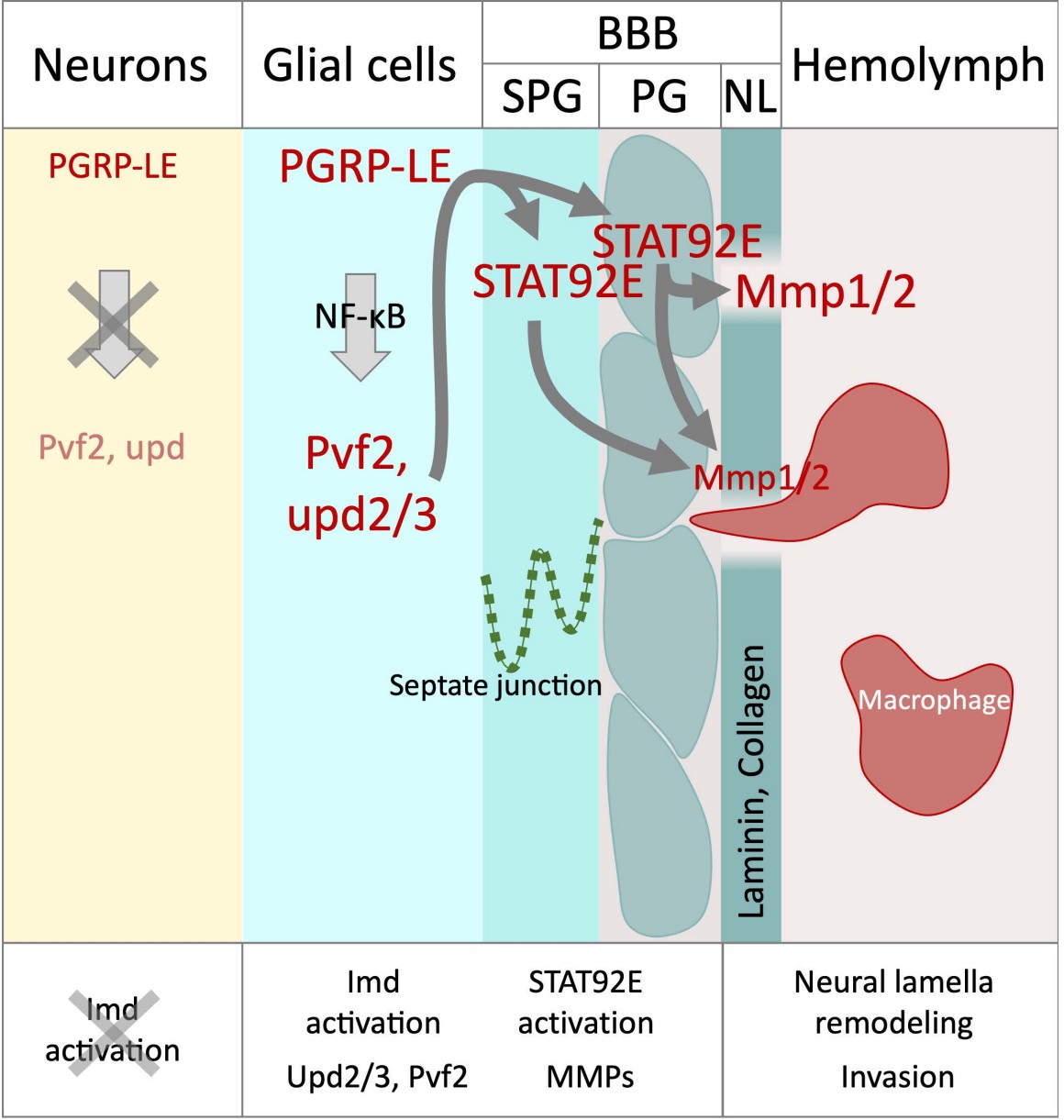

**Fig 10. Proposed model for signaling cascade upon panglial immunity induction.** Panglial expression of *PGRP-LE* activates the Imd pathway resulting in the activation of the NF-κB transcription factor Relish. Upon activation, JAK/STAT cytokines *upd2* and *upd3* are upregulated together with the *PDGF-/VEGF-related factor Pvf2*. Both Upd cytokines activate JAK/STAT signaling in glial cells of the blood–brain barrier. Activated JAK/STAT signaling triggers the expression of *Mmp1* and *Mmp2*. Neural lamella remodeling together with CNS *Pvf2* expression then triggers a pupal macrophage invasion into the CNS.

by hormonal signals to be more migratory and phagocytic as during this stage ecdysone-mediated disintegration of larval tissue starts, including that of the gut [60]. Alternatively, active remodeling of the neural lamella which normally takes place during early pupal stages sets the entry point to a successful invasion of macrophages. The results obtained in this study are in favor of this model as, for example, premature remodeling of the neural lamella by forced expression of *MMPs* enables an earlier invasion of the brain already during larval

stages. Interestingly, the two genes encoding MMPs in *Drosophila* show differential regulation upon immunity induction. This may be due to the differential functions of Mmp1/2. While Mmp1 is primarily a collagenase, but also targets membrane bound receptors, Mmp2 can act as a gelatinase. For example, during fat body remodeling during metamorphosis, Mmp1 preferentially cleaves DE-cadherin junctions between individual fat cells, while Mmp2 preferentially degrades the surrounding ECM [61]. Thus, both MMPs likely act in a sequential and cooperative manner.

The relevance of ECM remodeling is similarly noted in invasive migration of tumor cells. In vertebrates, tumor cells can upregulate *MMP2* and *MMP9* for ECM breakdown of the surrounding tissue in order to facilitate tumor progression and metastasis [62]. Here, specifically STAT3 can activate the expression of MMPs either in a JNK-dependent or MAPK-independent manner [48,63]. Additionally, STAT3 has been reported to act downstream of NF-κB in reactive astrocytes within the brain parenchyma [64]. Likewise, we report here that STAT92E is downstream of PGRP-LE which acts in a NF-κB-dependent manner. This suggests similarities between different immunity responses.

In *Drosophila,* JAK/STAT signaling is mostly active in the BBB and is needed for efficient macrophage invasion. Surprisingly, we found that mutations of the *Drosophila* cytokines Upd2 and Upd3 cause an invasion of macrophages into the nervous system even in the absence of induction of an immune response. This might suggest that a certain level of JAK/STAT signaling and MMP activity is required for an intact BBB, as suggested for the vertebrate BBB [5].

In mammals, invasion of peripheral immune cells into the CNS parenchyma involves two sequential and differentially regulated steps. First, immune cells must cross the endothelial BBB. This process is initiated by rolling and adhesion of immune cells to endothelial VCAM-1 and ICAM-1 via integrins, allowing an interaction of immune cells with chemokines expressed on endothelial cells [65–67]. Eventually, paracellular diapedesis of immune cells across endothelial tight junctions is initiated [68,69]. Immune cells accumulate within the perivascular cuff, the space between the endothelial BM and the perivascular BM of the glia limitans [9]. In *Drosophila*, immune cells also have to invade the CNS in two differentially regulated steps. However, here they first must overcome the neural lamella and subsequently invade the CNS over the BBB. We do not know if macrophage diapedesis at the *Drosophila* BBB happens in a transcellular or paracellular manner and if they preferentially invade at tricellular septate junctions.

As a second step of invasion in vertebrates, immune cells must overcome the glia limitans, which is prohibited under normal physiological conditions [70]. In the mammalian nervous system, expression of *MMP-2* as well as of *MMP-9* is upregulated in astrocytes of the glia limitans and is essential to allow an escape of peripheral immune cells into the CNS parenchyma [13]. In addition to promoting a chemotactic gradient by cleavage of local chemokines, these MMPs also induce the cleavage of β-dystroglycan which anchors astrocytic endfeet to the parenchymal BM, leading to as yet undefined changes in cell matrix interactions at the site of leukocyte entry [9]. There is no evidence of the broad ECM loss in vertebrates that is observed in *Drosophila*; however, MMP-2 and MMP-9 also cleave Notch-1 on astrocytic endfeet which promotes alleviation of PTEN suppression on AKT/NF-κB pathways [9,13]. Thereby, astrocytes upregulate chemokine expression and secretion, which promotes immune cell migration across the parenchymal border. Whether JAK/STAT signaling is also involved in leukocyte penetration of the parenchymal border is not clear and requires investigation.

Our data demonstrate that in the *Drosophila* system JAK/STAT signaling induces *MMP* expression in glial cells of the BBB. Here, MMPs are required for the remodeling of the

neural lamella, a dense ECM. Interestingly, both dystroglycan as well as Notch are specifically expressed by the *Drosophila* BBB, however, their relation to MMP activity is not known yet. Dystroglycan is known to bind ECM proteins specific for the parenchymal BM in vertebrates, i.e., laminins 111 and 211 [9,13], and it is required for the organization of these laminin isoforms and BM heparan sulfate proteoglycans in BM assembly [71]. During embryonic development, *Drosophila* macrophages migrate across tissue barriers in a defined time window [72]. Here, disintegration of focal adhesion was shown to facilitate invasive migration of macrophages [73,74]. Hence, cleavage of dystroglycan at the BBB by MMPs could disrupt anchorage of PGs to the neural lamella and, thereby, facilitate macrophage invasion in our model.

Together, our findings suggest a role for JAK/STAT signaling in regulating and balancing immune responses in the *Drosophila* CNS. Further studies using our model might help to elucidate the impact of JAK/STAT signaling on BBB integrity including the involvement in macrophage invasion over the BBB.

## Materials and methods

All experiments were conducted according to the regulations of the German legislation. An approval by an ethics committee is not required for *Drosophila* experiments.

### Fly work

All flies were raised according to standard procedures at 25 °C unless otherwise noted. Crosses including a *tub-P-Gal80^{ts}* were raised at 18 °C and shifted at the indicated time to 29 °C to allow Gal4-directed expression. The *QUAS-Pvf2* transgene generated in this study was inserted into the landing site attP40 using standard phiC31-integration protocols [75].

The following flies were obtained from public stock centers: *w[*]; P{w[+mC] = UAS-PGRP-LE.FLAG}2* (BDSC 33054), *w[1118]; P{w[+m*] = GAL4}repo/TM3, Sb[1]* (BDSC 7415), *y[1] w[*]; P{w[+mC] = PTT-un}vkg[G00454]* (BDSC 98434), *w[*]; P{w[+mC] = UAS-Timp.P}3* (BDSC 58708), *y[1] v[1]; P{y[+t7.7] v[+t1.8] = TRiP.JF01336}attP2* (BDSC 31489), *y[1] v[1]; P{y[+t7.7] v[+t1.8] = TRiP.JF01337}attP2* (BDSC 31371), *y[1] sc[*] v[1] sev[21]; P{y[+t7.7] v[+t1.8] = TRiP.HMC03539}attP2* (BDSC 53310), *w[1118]; P{w[+mC] = 10XStat92E-DGFP}2/CyO* (BDSC 26199), *y[1] v[1]; P{y[+t7.7] v[+t1.8] = TRiP.HMS00035}attP2* (BDSC 33637), *y[1] sc[*] v[1] sev[21]; P{y[+t7.7] v[+t1.8] = TRiP.HMS01293}attP2* (BDSC 34618), *y[1] sc[*] v[1] sev[21]; P{y[+t7.7] v[+t1.8] = TRiP.GL00305}attP2* (BDSC 35386), *w[*] upd2[Delta]* (BDSC 55727), *w[*] upd2[Delta] upd3[Delta]* (BDSC 55729), *w[1118] upd3[Delta9]; sna[Sco]/CyO* (BDSC 98419), *w[1118]; P{w[+mC] = UAS-GFP.dsRNA.R}143* (BDSC 9331), *y[1] w[*]; betaTub60D[Pin-1]/CyO; P{w[+m*] = ET-QF2.GU}repo/TM6B, Tb[1]* (BDSC 66477), *w[*]; P{y[+t7.7] w[+mC] = 10XUAS-mCD8-GFP}attP2* (BDSC 32184), *w; UAS-LacZ.nls* (BDSC 3955), *w[*]; P{w[+mC] = tubP-GAL80[ts]}10; TM2/TM6B, Tb[1]* (BDSC 7108), *w[1118]; P{w[+mC] = XP}Pvf2[d02444]* (BDSC 19631), *y[1] w[67c23]; Mi{PT-GFSTF.2}Mmp2[MI00489-GFSTF.2]/SM6a* (BDSC 60512). All other fly stocks were provided by other labs or were generated: *Mdr65-mtd-Tomato* (Gift from E. Contreras), *Sp/CyO; srpHemo-H2A::3xmCherry* (Gift from D. Siekhaus) [55], *Sp/CyO; srpHemo-moe::3xmCherry* (Gift from D. Siekhaus) [55], *UAS-MMP2, UAS-MMP1.2* [42], *QUAS-Pvf2* (this study), *2xaptE01-Gal4* (Gift from E. Contreras), *moody-Gal4* [19], *Puc::GFP* (Gift from M. Uhlirova), *UAS-LacZ^{dsRNA}* (Gift from S. Schirmeier), *w[*]; P{w[+mC] = UAS-Mmp1.f1}3* (BDSC 58701) [42], *vasPhiC31; attP40; attP2* (Gift from S. Luschnig)

## Immunohistochemistry

Immunohistochemistry for larval and pupal brains was performed using standard protocols [20]. For neural lamella and STAT92E::GFP staining during pupal stages, white prepupae were collected and kept in a moist chamber at 25 °C until dissection. Filet preparations were performed in phosphate-buffered saline by carefully cutting dorsally from the posterior to the anterior site avoiding to contact the brain using a Vannas scissors. Filets were subsequently fixed with Bouin's for 3 min followed by the standard staining protocol. The following antibodies were used: anti-dsRed (1:1,000, Takara), anti-mCherry (1:1,000; Invitrogen), anti-GFP (1:1,000; Abcam), anti-GFP (1:1,000; Invitrogen), anti-Laminin-γ (1:1,000; Abcam), anti-Repo (1:1,000; gift from B. Altenhein), anti-Sxl (1:1,000; Developmental Studies Hybridoma Bank), anti-MMP-1 (4 different monoclonal antibodies, supernatant 1:50, concentrate 1:500; Developmental Studies Hybridoma Bank), FluoTag X4 GFP (1:500, NanoTag Biotechnologies), FluoTag X4 RFP (1:500, NanoTag Biotechnologies), and anti-HRP DyLight 647 (1:500; Dianova). All secondary antibodies (Alexa Fluor 405, and Alexa Fluor 647 coupled; 1:500; Alexa Fluor 488 and Alexa Fluor 568, coupled; 1:1,000) were obtained from Invitrogen. All images were acquired using a Zeiss LSM 880.

## Quantification of macrophage invasion using arivis Vision4D

For quantification of the number of infiltrated macrophages, pupal CNS preparations were stained using standard protocols. Secondary antibodies were incubated over night at 4 °C and one additional hour at room temperature on the next day. Anti-Repo antibodies were used to label glial cells, anti-dsRed antibodies to label infiltrating macrophages, and anti-HRP to define the CNS. Sex-lethal staining was used to determine the sex of the pupae. Confocal images were taken using a 20× objective at a resolution of 512 × 512. Macrophage segmentation and counting was performed using arivis Vision 4D software (version 4.1.1; Zeiss, Jena, Germany).

An analysis pipeline was created, including a machine learning object classifier step to identify macrophages inside the CNS. First, the shape of the CNS was defined using the HRP signal. For object creation, the HRP signal is denoised with discrete gaussian method and a diameter of 10 μm (S6A Fig). The denoised HRP channel is then used for an intensity threshold segmenter with full connectivity in *X/Y* and a simple threshold of 7. The HRP objects were filtered with a threshold of volume > 6,000 μm³ (S6B Fig). Since the HRP signal strength can vary within a brain, the region growing operation was used with the watershed method, a threshold of 4 and max. distance of 140 μm. This results in one object, resembling the CNS (S6C Fig).

For segmentation of mCherry labeled macrophages, we used the blob finder operation with full connectivity in *X/Y* and a diameter of 3 μm, a probability threshold of 9.09 and a split sensitivity of 43,24% (S6D Fig). Subsequently, these objects were filtered using the segment feature filter operation. Macrophages were filtered with a threshold of a volume > 45 μm³. Since *srpHemo-H2A::3xmCherry* also labels some glial cells [20], Repo positive structures were also segmented. All objects with co-staining of mCherry and Repo were discarded. The remaining macrophage objects were then filtered for mean intensities, to remove objects with faint staining (threshold < 30) (S6E Fig).

Next, we calculated the distances of macrophages to the CNS object which were subsequently used for a Machine Learning Object classifier operation which was trained with the arivis Vision4D "Machine Learning Trainer (Objects)" tool. This defined macrophages inside the CNS (S6F Fig, red), or outside the CNS (S6F Fig, yellow). For training all channels were

used. Parameters include 3D oriented bounds, bounding box, bounding box (Pixel), center of bounding box, center of geometry, plane, center of mass (of objects defined for each color/excitation wavelength), hole statistics, projections (*XY/Z*), sphericity, surface area, volume, and calculated distances of macrophage nuclei and CNS object. After classification of macrophages, results are stored and were used for macrophage counting.

Importantly, macrophage nuclei which localize at the outermost glial layer of the CNS were not considered as infiltrated. Results of this analysis pipeline were manually validated and corrected for all the counting experiments.

## Quantification of the relative amount of macrophages using dot blots

Single wandering third instar larvae of the following genotypes were homogenized on ice-cold RIPA buffer [control: *repo-Gal4; UAS-nLacZ, srpHemo-H2A::3xmCherry*; Imd induction: *repo-Gal4; UAS-PGRP-LE, srpHemo-H2A::3xmCherry*; Imd induction with STAT92E knockdown: *repo-Gal4; UAS-PGRP-LE, UAS-STAT92E^{dsRNA}, srpHemo-H2A::3xmCherry*]. The protein extract was centrifuged for 10 min at 13,000 rpm and supernatant was kept at 95°C for 5 min. The protein extract of a single larvae was use for 6 dot blots (Schleicher and Schüll, Germany), three technical replicates were used for tubulin as normalization control and three technical replicates were used to determine the mCherry signal to deduce the relative macrophage amount. The following antibodies were used: anti-dsRed (1:1,000, Takara), anti-Tubulin (AA4.3-s, Developmental Studies Hybridoma Bank), anti-rabbit Peroxidase-conjugated (1:7,500, Jackson Immunoresearch) and anti-mouse Peroxidase-conjugated (1:7,500, Jackson Immunoresearch). Ten whole larvae were used per genotype.

## Molecular genetics

For generation of the QUAS-Pvf2 construct, the cDNA clone RH40211 (DGRC Stock 10757; https://dgrc.bio.indiana.edu//stock/10757; RRID:DGRC_10757) was cloned into the pQUAST-attB (Addgene plasmid # 104880; http://n2t.net/addgene:104880; RRID:Addgene_104880) by T4 ligation. For this, the cDNA clone as well as the entry plasmid were digested with Kpn I together with high fidelity Not I (both NEB) and subsequently ligated.

For quantification of relative expression levels, a qRT-PCR was performed. RNA was isolated from 15-17 third instar larval or 7–10 pupal brains using the RNeasy mini (QIAGEN) kit. cDNA was synthesized using QuantiTect (QIAGEN) kit according to the manufacturer's instructions. qRT-PCR was performed using a TaqMan gene expression assay in a qTower³ Real-Time PCR Thermal Cycler system (analytic jena; see below) together with a TaqMan Universal PCR Master Mix II (Life Technologies) (see Table 1). RpL32 was used as a housekeeping gene. For all samples, a minimum of four biological replicates were analyzed and sexes were separated.

**Table 1. TaqMan (Thermo Fisher) gene expression assays used for qRT-PCR experiments.**

| Gene | TaqMan gene expression assay |
|------|------------------------------|
| *Upd1* | Dm01843791_g1 |
| *Upd2* | Dm01844134_g1 |
| *Upd3* | Dm01844142_g1 |
| *Rpl32* | Dm02151827_g1 |
| *Mmp1* | Dm01820359_m1 |
| *Mmp2* | Dm01794359_m1 |

## Neural lamella visualization

Visualization of the neural lamella was performed with arivis Vision4D software (version 4.1.1, Zeiss, Jena Germany). For visualization of the integrity of the neural lamella, a surface was created by creating an iso-surface of the according channel stained for the neural lamella.

## Fluorescence quantification

To quantify the fluorescence using the different signaling reporter, larval brains were dissected, fixed in formaldehyde and subsequently kept in the dark until recording. Brains were always recorded at high resolution (16 bit) with the same laser intensities. Fluorescence intensity was determined with Fiji (ImageJ version 2.14.0/1.54f) [76]. The corrected total cell fluorescence (CTCF) was calculated in arbitrary units by measuring the integrated density of whole brains and subtracting the background signal (CTCF = integrated density − (area of brain * mean fluorescence of background).

## Quantification and statistical analysis

Statistical analysis and calculations were acquired using Excel and Prism 6.0. Normality was tested when samples were $n > 12$ by applying the Shapiro-Wilk-Test. For normal values, we performed an unpaired Student $t$ test, and for values not following a normal law, we chose Mann–Whitney. When more than two conditions were compared, non-parametric Kruskal–Wallis followed by Dunn's multiple comparisons test was performed, generating adjusted $p$-values (alpha = 0.05). $p$-values lower than 0.05 were considered significant.

## Supporting information

**S1 Fig. Tools used in this study.** Schematic representation of all main tools used in this study. *repo-Gal4* and *repo-QF2* direct expression in all glial cells. *moody-Gal4* directs expression in the subperineurial glia (SPG), *aptE01-Gal4* directs expression in the perineurial glia (PG). The neural lamella (NL) can be labelled using a GFP protein-trap insertion in the *collagen IV* gene or by staining using anti-Laminin-γ. Macrophages (Mø) can be visualized using *srpHemo-H2A::3xmCherry* or by *srpHemo-moe::3xmCherry*.
(TIFF)

**S2 Fig. Mmp1 upregulation is independent of JNK signaling.** (**A**, **B**) Third instar larval brains of control (A) and immunity induction (B) expressing the JNK reporter construct *puc::GFP*. Scale bar, 100 μm. (**C**) Measurement of JNK signaling activation in the CNS of third instar larvae with and without panglial immunity induction. No significant difference is detected. Mann–Whitney, $P = 0.5543$. Control $n = 16$, IMD $n = 24$. The corrected total cell fluorescence (CTCF) was calculated in arbitrary units (AU), see Materials and methods for details. For quantification see S8 Data. (**D**, **E**) Third instar larval brains with a panglial immunity induction (D) and simultaneous *basket* downregulation (E) stained for Mmp1. Note the Mmp1 signal at the surface of the CNS (D′, E′, arrowheads). Scale bar, 100 μm.
(TIFF)

**S3 Fig.** Inhibition of panglial JAK/STAT signaling delays neural lamella remodeling (**A–D**) Dissected brains of pupae carrying a JAK/STAT signaling reporter. The age and genotypes are indicated. Brains were stained for JAK/STAT signaling activation (green), HRP (blue) to label neuronal membranes and *mCherry* expression (magenta) directed by the macrophage marker *srpHemo-moe::3xmCherry*. Note the increased GFP signal upon a panglial immunity induction. Scale bars, 100 μm. (**E**, **F**) Pupal brains stained for Laminin-γ (green) and HRP (blue) upon a panglial *STAT92E* knockdown (E) and controls expressing *LacZ*$^{dsRNA}$. Neural lamella

remodeling is inhibited upon panglial *STAT92E* knockdown compared to controls. Laminin-γ surfaces were generated with arivis4D.
(TIFF)

**S4 Fig. Neural lamella remodeling upon panglial Pvf2 expression.** (**A**) Pupal brains with a panglial *Pvf2* expression stained for Laminin-γ (green), HRP (blue) and a macrophage marker (mCherry expression (magenta) directed by *srpHemo-moe::3xmCherry*). The neural lamella remodeling starts in 20 h APF pupal stages. (**B**, **C**) JAK/STAT signaling is not activated in pupal stages upon a panglial *Pvf2* expression. Pupal brains expressing *Pvf2* and a JAK/STAT signaling reporter of indicated ages stained for STAT92E activation (green), HRP (blue) a macrophage marker (mCherry expression (magenta) directed by *srpHemo-moe::3xmCherry*). (**D**) Pupal CNS expressing *Pvf2* in all glial cells stained with repo (green) to label glial nuclei, HRP (blue) to label neuronal membranes, and dsRed (magenta) directed by the macrophage marker *srpHemo-moe::3xmCherry*. Many macrophages are located within the CNS. Scale bars, 100 μm.
(TIFF)

**S5 Fig. Macrophages are attached to brain following Mmp2 expression in SPGs.** (**A**) Larval CNS with a panglial QUAS-directed *Pvf2* expression and Gal4 induced *Mmp1* expression in the subperineurial glia. *Mmp1* expression was restricted to late larval stages by a ubiquitously expressed *Gal80^ts* and an appropriate temperature regime. Brains were stained for Laminin-γ (green), mCherry expressing macrophages (magenta), and HRP (blue) to label neuronal membranes. The neural lamella appears rather intact, and macrophages never invaded the larval CNS (A′, asterisk). $n$ = 19. Scale bar, 100 μm. (**B**) Upon expression of *Mmp2* macrophages breach the neural lamella and are firmly attached to the CNS (B′, B″, arrowheads). The neural lamella appears disrupted (B″, asterisk). $n$ = 5. Scale bar, 100 μm.
(TIFF)

**S6 Fig. arivis Vision4D pipeline for macrophage counting.** Quantification of infiltrated macrophages was performed with arivis Vision 4D. (**A**) The HRP staining of a pupal CNS used for segmenting the CNS. First, the HRP signal was denoised by discrete gaussian. (**B**) Same CNS, anti-Repo and anti dsRed staining to show glial nuclei (green) and macrophages (red). (**C**) In order to fill regions within the CNS with low-HRP staining signal, the CNS object was enlarged by region growing and later used for defining infiltrated macrophages by machine learning. (**D**) Objects expressing the nuclear macrophage marker *srpHemo-H2A::3xmCherry* were created using the blob finder. (**E**) These objects were filtered against co-localization with Repo (B) resulting in segmented macrophage nuclei. (**F**) Subsequently, these objects were classified with machine learning into macrophages inside (magenta) and outside (yellow) the CNS.
(TIFF)

**S1 Raw Images. Raw data for quantification of macrophage numbers.** The loading scheme and six dot blots performed to quantify the number of macrophages in third instar larvae of the indicated genotypes are shown. Dot blots stained with the antibody indicated. Each sample (one larva) was applied as three technical replicates. Samples excluded from the analysis are indicated by a cross.
(PDF)

**S1 Data. Ct values and quantification of relative expression levels of Mmp1, Mmp2 (gene of interest, GOI), and Rpl32 (housekeeping gene, HKG) of controls and immunity induction of three different developmental stages (third instar wandering larvae (wL3), 8 h APF and 16 h APF).**
(XLSX)

**S2 Data. Number of invaded macrophages of the indicated genotypes quantified by using the arivis4D pipeline.**
(XLSX)

**S3 Data. Measurement and quantification of JAK/STAT activation of controls and during immunity induction of whole brains of third instar wandering larvae.**
(XLSX)

**S4 Data.** Tables 5C, 5D: Number of invaded macrophages of the indicated genotypes quantified by using the arivis4D pipeline. Tab 5D: Measurement and quantification of the dot blots shown in S1 Raw Images.
(XLSX)

**S5 Data. Ct values and quantification of relative expression levels of Mmp1, Mmp2 (gene of interest, GOI), and Rpl32 (housekeeping gene, HKG) of controls, immunity induction and immunity induction with concomitant STAT92E knockdown of two different developmental stages (third instar wandering larvae (wL3) and 8 h APF).**
(XLSX)

**S6 Data.** Table 7A: Ct values and quantification of relative expression levels of upd1, upd2, upd3 (gene of interest, GOI), and Rpl32 (housekeeping gene, HKG) of controls and immunity induction of three different developmental stages (third instar wandering larvae (wL3), 8 h APF and 16 h APF). Table 7B: Number of invaded macrophages of the indicated genotypes quantified by using the arivis4D pipeline.
(XLSX)

**S7 Data. Number of invaded macrophages of the indicated genotypes quantified by using the arivis4D pipeline.** Moreover, the measurement and quantification of JAK/STAT activation of controls and panglial Pvf2 expression of whole brains of third instar wandering larvae is shown.
(XLSX)

**S8 Data. Measurement and quantification of JNK activation of controls and during immunity induction of whole brains of third instar wandering larvae.**
(XLSX)

## Acknowledgments

We are indebted to B. Altenhein, E. Contreras, S. Luschnig, S. Schirmeier, D. Siekhaus, and M. Uhlirova for sharing reagents. We thank Lydia Sorokin and members of the Klämbt lab for critical reading of the manuscript and help throughout the project.

## Author contributions

**Conceptualization:** Bente Winkler, Christian Klämbt.

**Data curation:** Bente Winkler.

**Formal analysis:** Bente Winkler.

**Funding acquisition:** Christian Klämbt.

**Investigation:** Bente Winkler.

**Methodology:** Bente Winkler, Dominik Funke.

**Project administration:** Bente Winkler.

**Resources:** Christian Klämbt.

**Software:** Bente Winkler.

**Supervision:** Christian Klämbt.

**Validation:** Bente Winkler, Dominik Funke, Christian Klämbt.

**Visualization:** Bente Winkler, Dominik Funke, Christian Klämbt.

**Writing – original draft:** Bente Winkler.

**Writing – review & editing:** Bente Winkler, Christian Klämbt.

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
