## [Editor Report · Decision Letter 0]

24 Jun 2024

Dear Christian, 

Great to hear from you!

Thank you for submitting your manuscript entitled "Macrophage invasion into the Drosophila brain requires JAK/STAT dependent MMP activation in the blood-brain barrier" for consideration as a Research Article by PLOS Biology.

Your manuscript has now been evaluated by the PLOS Biology editorial staff as well as by an academic editor with relevant expertise and I am writing to let you know that we would like to send your submission out for external peer review.

Once your full submission is complete, your paper will undergo a series of checks in preparation for peer review. After your manuscript has passed the checks it will be sent out for review. To provide the metadata for your submission, please Login to Editorial Manager (https://www.editorialmanager.com/pbiology) within two working days, i.e. by Jun 26 2024 11:59PM.

Kind regards,

Christian

Christian Schnell, PhD

Senior Editor

PLOS Biology

cschnell@plos.org

---

## [Decision Letter · Decision Letter 1]

22 Aug 2024

Dear Christian,

Thank you for your patience while your manuscript "Macrophage invasion into the Drosophila brain requires JAK/STAT dependent MMP activation in the blood-brain barrier" was peer-reviewed at PLOS Biology. Apologies for the delay in getting back to you, but it took a bit longer than I hoped to find reviewers during the holiday time. Your manuscript has now been evaluated by the PLOS Biology editors, an Academic Editor with relevant expertise, and by several independent reviewers. 

In light of the reviews, which you will find at the end of this email, we would like to invite you to revise the work to thoroughly address the reviewers' reports.

As you will see below, the reviewers find the study very interesting and overall think that the study is very well executed and provides important insights. However, they also raise a couple of concerns that need to be addressed for publication.

Given the extent of revision needed, we cannot make a decision about publication until we have seen the revised manuscript and your response to the reviewers' comments. Your revised manuscript is likely to be sent for further evaluation by all or a subset of the reviewers.

**IMPORTANT - SUBMITTING YOUR REVISION**

*Re-submission Checklist*

*Published Peer Review*

*PLOS Data Policy*

*Blot and Gel Data Policy*

Sincerely,

Christian

Christian Schnell, PhD

Senior Editor

PLOS Biology

cschnell@plos.org

REVIEWS:

Reviewer #1: Winkler and Klambt investigate the mechanism behind macrophage invasion into the brains during neuroinflammation. This manuscript is a continuation of a previous study by the same group that demonstrated neuroinflammation triggers infiltration of macrophages across the blood-brain barrier during brain development in Drosophila.

This new manuscript shows that the immune response triggered by activation of the Imd pathway leads to the upregulation of UPD, activation of the JAK/STAT signaling pathway in the glial cells of the BBB. This activation results in the expression of MMPs, which are necessary for the remodeling of the neural lamella surrounding the fly brain. The remodeling of the ECM allows for the macrophages to invade the CNS. This study contributes to our understanding of JAK/STAT signaling in neuroinflammation.

The manuscript is well-written and experiments are logical. The authors make use of the well-established glial-specific GAL4 to genetically manipulate IMD, MMP, and JAK/STAT signaling in a tissue-specific manner. They use immunohistochemistry of the larval brain to determine changes in neural lamina and macrophage invasion. The conclusions are generally well supported by the evidence. The only weakness is that the macrophage invasion quantification is not well defined and may need an additional control (see below for suggestions.

Specific Comments/suggestions:

1. The method of quantifying macrophage number is unclear as described. Is it possible that the genetic interventions of using dsSTAT/ and other genetic manipulations resulted in overall decrease in macrophage number surrounding the brain? It would be helpful to have infiltrated macrophage numbers you presented as a ratio over total macrophages observed in your image (or number of macrophages in the hemolymph).

2. Figure 6A image stitching looks to be abnormal.

3. Number of experimental replicates are not mentioned.

4. n's are missing in Figure 1.

5. How many larval brains were used for the qRT-PCR experiments?

6. Could the authors clarify the reasoning behind using 2^-dCT rather than the typical representation as fold change over biological control (2^-ddCT)?

7. Figure 4D - STAT::GFP expression does not appear to overlap with mdr65 expressing cells at the center of the VNC (line of small circles in the middle). Could you speculate on why this is?

8. Line 46-48: while endothelial cells do produce components of the basement membrane, it might be more balanced to mention the involvement of pericytes as well (https://doi.org/10.1038/srep36450; https://doi.org/10.3390/cells11101707).

9. A few discussion and conclusion points could be clarified - the authors mention that the infiltration of immune cells can be a physiological or pathological change, however the idea is very general. My question is, in your model, until what point would an infiltration of immune cells associated with inflammation be considered necessary for physiological processes and at what point would it be considered pathological?

10. Does the infiltration of immune cells associated with a neuroinflammatory event during the pupal stage of Drosophila impact the development of the fly or adult behavior and neurological phenotypes?

Reviewer #2: The manuscript "Macrophage invasion into the Drosophila brain 1 requires JAK/STAT2 dependent MMP activation in the blood-brain barrier" by Winkler and Klämbt use Drosophila melanogaster to identify molecular mechanisms involved in the invasion of immune cells into the brain. The authors demonstrate that immune cell invasion of the fly brain requires inflammatory cytokines that activate the JAK/STAT pathway which promotes the expression of MMPs and the remodeling of the extracellular matrix. The authors conclude that this mechanism may be conserved in vertebrates and could be useful in understanding the role of JAK/STAT during neuroinflammation. 

This manuscript is well written, but requires some clarification to non-specialists, especially to those not well versed in Drosophila genetics (please see major concern #1). The experiments are logical and the data support the claims by the authors, although some control experiments are needed for some of the results. Overall, this reviewer finds the study very interesting and significant given that neuroinflammation plays a prominent role in many neurological diseases. While the applicability from invertebrates to vertebrates may not be relevant for all of the results presented, the manuscript provides valuable information that could be translatable to human studies.

Major concerns:

1) This reviewer found the genetic lines used very difficult to understand initially. For example, in several sections of the results, it was challenging to follow the experiments being performed. This reviewer suggests providing a detailed description of the lines used and the rationale behind each line so that the reader does not have to spend too much time trying to figure out what was done. For example, repo-Gal4 was not described anywhere in the manuscript. To determine that this line expresses in glial cells, this reviewer had to refer to Winkler et al Sci Adv 2021. While some of this information is provided in the legends, a description in the main text would have been very useful. Perhaps a figure with an overview of the experimental lines and structures as shown in Winkler et al Sci Adv 2021 could help clarify these important points.

2) Knockdown of STAT, dome, and hop in results section "JAK/STAT signaling is required during macrophage invasion" shows a dramatic decrease in the number of infiltrating macrophages. However, this data does not distinguish between reduced infiltration versus a reduced total number of macrophages. This could be demonstrated by showing that the number of macrophages in the periphery is not impacted by knockdown of one or more of these genes. 

3) The analysis of upd gene expression was performed on CNS preparations. However, it seems likely that immunity induction could also regulate expression outside of the CNS, impacting BBB permeability and macrophage infiltration. Along these lines, the study used upd2 and upd3 knockouts as opposed to glia-specific dsRNA for knockdown. With dsRNA, the CNS effects of upd1 could potentially be examined as well. Ultimately, the question is: do Unpaired cytokines translocate across the Drosophila BBB and is this transport bidirectional? For example, the author's model (Fig. 10) shows the possibility of Pvf2 moving across the BBB.

Additional comments:

1) In Figure 2, Mmp1 staining is shown to be localized to the Drosophila BBB. However, there are no co-localized markers demonstrating that this staining represents the BBB (i.e. what cell type(s) show that this structure is the BBB as shown in Fig 4 for example?). Without significant familiarity with the Drosophila BBB, this information would be useful to put the results in context. 

2) Figure 3A and 3B are lacking controls for "No Immunity Induction." This would be useful to compare to TIMP and MmpdsRNA. In other words, does TIMP restore macrophages to "no immunity induction" levels and does MmpdsRNA reduce macrophages from baseline? Also, HRP is not described (Assuming that this is neuronal membrane as described in Figure 1 legend?).

3) In Figure S2A, CTCF is not defined. According to the text, it could be assumed that this is the level of puc::GFP expression, but is not described in the figure legend. 

Furthermore, it would be relevant to provide images of puc::GFP expression to demonstrate that localization of the signal does not change upon IMD. Likewise, knockdown of basket appears to unexpectedly upregulate Mmp1 compared to control (Fig. S2B,C), although this data is not quantified.

4) Please explain how STAT92E being downregulated. For example, in Fig. 5A, is STATdsRNA actually UAS::STAT92EdsRNA? This is not described in the text or methods or at least not easily found.

5) The JAK/STAT signaling seems to contribute to the level of Mmp expression (Fig. 6). However, these data do not show statistical significance. As n=4 for this analysis and there appears to be a wide range of expression between data points, perhaps increasing the numbers would provide significance. This also seems to be the case for expression of the upd genes (Fig. 7). Please elaborate as to why more samples were not included in these analyses.

6) There are several concepts throughout the manuscript that cite review papers rather than the primary literature. Please consider modifying the references accordingly.

Reviewer #3: This manuscript by Winkler et al attempts to unravel the signalling pathways driving immune cell invasion into the brain during neuroinflammation using Drosophila as a model system. The manuscript follows up from previous work from the lab which established a neuroinflammation model in drosophila to study macrophage invasion into brain. The current work identifies a role for unpaired cytokines in activating JAK/STAT signalling pathway to drive changes in ECM remodelling allowing macrophage invasion. The identification of the molecular pathway is an advance on the previous work and demonstrates interesting parallels between the vertebrate and invertebrate neuroinflammatory pathways. A limitation of the work is that while the players are identified, the mechanistic detail of how the pathways work is not present which makes the manuscript rather descriptive. It is not clear how the two MMPs are differentially activated ( is there a cell type specific requirement?). Also, the evidence supporting the differential expression of JAK/STAT signalling in the two cell types is rather weak.

1. The structure of the brain in the pupal stages look different in the inflammation model as compared to the control. Is there a structural degradation of the brain ? Does that happen before the loss of the ECM? The authors could supply transmitted light images to make this clearer.

2. The authors do not discuss in any detail why the MMP expression patterns are quite temporally different. The levels of MMP1 in the inflammation model is high in the larval stages and stays up till 8APF and then drastically goes down. The MMP2 levels only increase at 8h APF but and then goes down. Interestingly the Laminin levels go down at 8 h APF and then consistently stays down. It would be interesting to understand why the ECM remodelling is temporally different from the MMP expression and what contributes to the differential MMP expression. Interestingly, loss of both the MMPs using RNAi has the same phenotype on Laminin. Also, can the authors use an antibody against MMP2 to see the expression pattern ?

3. A technical concern within the manuscript is the interchangeable use of both the larval brains and the pupal stages for different analysis. For example, the STAT reporter expression is only analysed in the WL3 stages and not the pupal stages. Does the STAT signalling go down later, leading to a lower level of MMP induction? 

4. The evidence of differential STAT reporter expression in the two glial cell types is weak. From the images in Figure 4D, it is not clear how the authors are distinguishing the two cell types. The images provided in D'-D''' look like a side view with the STAT reporter expressed on both the apical membranes and the cortex of the same cell which expresses Ds-red. The authors should distinguish the two cell types in the images to provide evidence of differential expression.

5. A major concern within the manuscript is the lack of information surrounding the automated workflow used to quantify the invaded macrophages. The authors should provide more details on how this was done, how was the macrophage assigned as invaded. Would the macrophages on the surface be counted? Images of the invaded macrophages could be added to the supplementary data.

Technical concerns:

1. The authors should provide cartoons to better indicate the different areas of the brain and the cells types to readers who are not familiar with the field.

2. There is no quantification of the laminin expression and since that is used as the major readout of ECM remodelling, addition of this quantification would be essential to provide a more quantitative description of the ECM levels. For example, calculating the fluorescence intensity for Laminin::GFP expression in figure 1 under wild type and under induced immunity conditions would provide clearer evidence that Laminin::GFP expression reduces over time and that this process is quickened when an immune response is induced. This same principle can be applied to figures 2, 3 and 4.

3. The authors describe that the expression of GFP tagged Collagen affects Laminin levels. This is a surprising finding and while the use of Laminin as a proxy for ECM remodelling is fine, it would be interesting to understand why the expression of the Collagen transgene has this effect.

4. Are the dsRNAi used in the manuscript (for the MMPs) previously described? It would be good if the authors can use more than one RNAi line for the proteins to negate any off-target effects.

Minor concerns:

1. The signalling pathways described from page 5, line 94 could be clearer to the reader if presented with a schematic diagram. 

2. The CollagenIV::GFP data in supplementary figure 1 could be moved to figure 1 as this is a key point describing why the authors focus on laminin throughout the paper.

3. In figure 3, a figure depicting what the authors defined as successful macrophage invasion would help the reader visualise where the macrophages are invading. A schematic accompanied with a microscopy image would work well. 

4. In figure 6, the axis for the graphs in C and D differ from those in figure 2 making it difficult for the reader to compare the two. 

5. Some of the figure legends lacked detail. The legend of figure 1 contains details that could be made simpler.

6. One of the panels in supplementary figure 5 looks out of focus making it difficult to compare it to its control.

7. 'Pupariation' is spelt incorrectly in the figure legend for figure 1. 

8. In figure 4, the image titles are overlapping with the CNS images.

---

## [Editor Report · Decision Letter 2]

13 Jan 2025

Dear Christian,

Thank you for your patience while we considered your revised manuscript "Macrophage invasion into the Drosophila brain requires JAK/STAT dependent MMP activation in the blood-brain barrier" for publication as a Research Article at PLOS Biology. This revised version of your manuscript has been evaluated by the PLOS Biology editors and the Academic Editor.

Based on our Academic Editor's assessment of your revision, we are likely to accept this manuscript for publication, provided you satisfactorily address the following data and other policy-related requests:

* We would like to suggest a minor correction to the title: 

Macrophage invasion into the Drosophila brain requires JAK/STAT-dependent MMP activation in the blood-brain barrier

* Please add the links to the funding agencies in the Financial Disclosure statement in the manuscript details.

* DATA POLICY:

Regardless of the method selected, please ensure that you provide the individual numerical values that underlie the summary data displayed in the following figure panels as they are essential for readers to assess your analysis and to reproduce it: 2AB, 3AB, 4C, 5CDE, 6CD, 7AB, 8AB and S2C.

* CODE POLICY

We expect to receive your revised manuscript within two weeks. 

*Published Peer Review History*

*Press*

Sincerely,

Christian

Christian Schnell, PhD

Senior Editor

cschnell@plos.org

PLOS Biology

---

## [Editor Report · Decision Letter 3]

24 Jan 2025

Dear Christian,

Thank you for the submission of your revised Research Article "Macrophage invasion into the Drosophila brain requires JAK/STAT-dependent MMP activation in the blood-brain barrier" for publication in PLOS Biology. On behalf of my colleagues and the Academic Editor, Richard Daneman, I am pleased to say that we can in principle accept your manuscript for publication, provided you address any remaining formatting and reporting issues. These will be detailed in an email you should receive within 2-3 business days from our colleagues in the journal operations team; no action is required from you until then. Please note that we will not be able to formally accept your manuscript and schedule it for publication until you have completed any requested changes.

PRESS

Sincerely, 

Christian

Christian Schnell, PhD

Senior Editor

PLOS Biology

cschnell@plos.org